# Cellular senescence-associated genes in rheumatoid arthritis: Identification and functional analysis

**You Ao, Qing Lan, Tianhua Yu, Zhichao Wang, Jing Zhang** [ORCID]*

Department of Orthopaedics, The Fifth Hospital of Harbin, Harbin, Heilongjiang, P. R. China

* ZhangJing_900411@163.com

**Data Availability Statement:** All data are in the manuscript and supporting information files

**Funding:** The author(s) received no specific funding for this work.

## Abstract

Rheumatoid arthritis (RA), a long-term autoinflammatory condition causing joint damage and deformities, involves a multifaceted pathogenesis with genetic, epigenetic, and immune factors, including early immune aging. However, its precise cause remains elusive. Cellular senescence, a hallmark of aging marked by a permanent halt in cell division due to damage and stress, is crucial in aging and related diseases. In our study, we analyzed RA microarray data from the Gene Expression Omnibus (GEO) and focused on cellular senescence genes from the CellAge database. We started by selecting five RA datasets from GEO. Next, we pinpointed 29 differentially expressed genes (DEGs) linked to cellular senescence in RA, aligning them with genes from CellAge. We explored the roles of these DEGs in cellular senescence through Gene Ontology (GO) and Kyoto Encyclopedia of Genes and Genomes (KEGG) pathway analysis. We then pinpointed three key genes (DHX9, CYR61, and ITGB) using random forest and LASSO Cox regression machine learning techniques. An integrated diagnostic model was created using these genes. We also examined the variance in immune cell infiltration and immune checkpoint gene expression between RA and normal samples. Our methodology's predictive accuracy was confirmed in external validation cohorts. Subsequently, RA samples were classified into three distinct subgroups based on the cellular senescence-associated DEGs, and we compared their immune landscapes. Our findings reveal a significant impact of cellular senescence-related DEGs on immune cell infiltration in RA samples. Hence, a deeper understanding of cellular senescence in RA could offer new perspectives for diagnosis and treatment.

## Introduction

Rheumatoid arthritis (RA) is a well-known autoimmune disease distinguished by chronic inflammation affecting many parts of the synovial tissue, particularly cartilage and bone [1,2]. Patients with RA experience symmetric, painful and incapacitating joint inflammation several years after the immune system has made crucial errors. Gradually, the rheumatoid pannus can lead to irreparable injury to tendons, cartilage, and bone[3]. However, the exact etiology is unclear. The incidence of RA varies around the world, with higher rates typically found in

**Competing interests:** The authors have declared that no competing interests exist.

developed countries. This might be due to exposures to environmental risk factors, genetic factors, demographic variations and inadequate reporting in other regions of the world [3]. A more comprehensive comprehension of the natural RA history, together with the contributing factors to the development of this disease in various populations, could potentially pave the way for introducing effective prevention strategies for this incapacitating condition.

Although the pathogenesis of RA remains unknown, the preponderance findings from genetics, tissue analysis, modeling, and clinical studies suggest an immune-mediated aetiology with stromal tissue dysregulation, which promote prolonged inflammation and joint damage [4]. And accumulating studies indicated that the synovial cavity of RA patients is highly enriched in dysfunctional immune cells, including T cells, B cells, macrophages, etc [5]. The early self-tolerance deficiency of RA is mainly related to the alteration in T cell homeostasis and studies have shown that the HLA-class II allele has been found to be the most important risk factor for this disorder, as it exacerbates T cell malfunctions [6]. Thus, the mentioned disease-related polymorphisms point to the T cell as being a crucial factor in the progression of RA. Nevertheless, the existence of autoantibodies prior to the disease onset and the effectiveness of B-cell depletion therapies imply that B cells also contribute significantly to the RA development and progression [4]. A fully understanding of the impact of immune cells in the development of RA will provide new insights into pathogenesis, leading to improved therapies and quality of life for patients.

The prevalence of RA and its associated complications is predicted to rise with increasing age. It is noteworthy that RA patients exhibit signs of accelerated senescence, especially in immunesenescence [7]. Senescence is a widespread biological phenomenon with various functions in tissue remodeling. It is also linked to significant modification in immune cells, leading to heightened vulnerability to infections, inadequate vaccine responses, susceptibility to autoimmunity and malignant growth [8]. Therefore, it seems plausible that immune senescence may contribute to autoimmunity-mediated diseases. In addition, Kalayjian et al. conducted a cross-sectional study on individuals living with HIV infection that initially demonstrated the correlation between immune senescence and T cell defects [9]. This phenomenon mainly affects T cells, which are subjected to considerable proliferative pressure in order to achieve clonal expansion and T cell proliferation homeostasis [10]. T cells of older adults exhibit genetic and epigenetic alterations, impaired mitochondrial fitness and failure to maintain proteostasis. These changes lead to a deviation from protecting to damaging the host[11].

Cellular senescence is an aging mechanism that is characterized by the cessation of cell division in the face of cellular injury and stress [12]. Senescent cells may undergo a distinct pathogenical aging-related secretory phenotype that stimulates secondary senescence and perturbs tissue homogeneity, thereby increasing the susceptibility of several chronic diseases [13]. In our research, we sourced RA datasets from the GEO database and genes linked to cell senescence from the CellAge database. Our initial step was to identify DEGs associated with cellular senescence. We then employed GO and KEGG analyses to delve into the biological processes of these senescence-related DEGs. Following this, we used random forest analysis to further pinpoint significant DEGs connected to senescence. Additionally, we utilized LASSO Cox regression analysis to select pivotal genes associated with cellular senescence in RA, leading to the development of the CSscore predictive model. Our study also included an examination of differences in immune cell infiltration between RA and normal samples. The effectiveness of our methodology was further confirmed through validation in external cohorts. Ultimately, we categorized RA samples into three distinct groups based on senescence-associated DEGs and conducted a comparative analysis of their immune landscapes.

## Materials and methods

### Data collection

Our study commenced with the acquisition of RA databases from the GEO database, accessed at (https://www.ncbi.nlm.nih.gov/geo/), using "Rheumatoid arthritis" as the search term. We set specific inclusion criteria for these datasets: they had to be from Homo sapiens, focus on expression profiling by array, and involve a minimum of 20 participants. This process resulted in the selection of five datasets: GSE55457, GSE12021, GSE55235, GSE77298, and GSE178557. Among these, GSE55457 was designated as the training dataset, with the others serving as validation datasets. The comprehensive details of these five cohorts are delineated in Table 1 of our study. In parallel, we also gathered data on 279 genes linked to cellular senescence from the Cell Age database, available at (https://genomics.senescence.info/cells/). The specifics of these genes are cataloged in S1 Table of our documentation.

### Data preprocessing and normalization

The RA cohorts were obtained from the GEO database and were subsequently processed and normalized using version 4.1.2 of the R statistical software (https://www.r-project.org/) and the Bioconductor analysis tools (http://www.bioconductor.org/). The "affy" R package was utilized to perform RMA background adjustment, complete log2-transformation, quantile normalization, and summarizing via the median-polishing algorithm. Probes lacking gene symbols were eliminated. For genes that were mapped to more than one prob, the final expression value was determined as the mean value of all probes.

### Identification of cellular senescence-related differentially expressed genes (DEGs)

In our study, the RA datasets sourced from the GEO database underwent processing and normalization using R statistical software version 4.1.2, accessible at https://www.r-project.org/, in conjunction with Bioconductor analysis tools, available at http://www.bioconductor.org/. We employed the "affy" package in R for RMA background correction, log2 transformation, quantile normalization, and summarization using the median-polishing algorithm. Probes lacking gene symbols were removed. For genes associated with multiple probes, we calculated the final expression value as the average of all probe values. To discern the biological differences between RA and control samples, we conducted differential expression analysis using the "limma" package, focusing on senescence-related genes. Genes meeting the criteria of P-value $< 0.05$ and $|\log2$ fold change (FC)$| > 0.5$ were classified as cellular senescence-related DEGs. The results were visualized in a volcano plot using the "ggplot2" package.

### Functional enrichment analysis of cellular senescence-related DEGs

For a deeper understanding of the biological mechanisms of cellular senescence-associated DEGs, we conducted GO and KEGG functional enrichment analyses using the

**Table 1. Summarizes the RA datasets included in our study.**

| Datasets | Training/Testing | No. patients | PMID |
|---|---|---|---|
| GSE55457 | Training | 33 | 24690414 |
| GSE12021 | Testing | 57 | 18721452 |
| GSE55235 | Testing | 30 | 24690414 |
| GSE77298 | Testing | 23 | 26711533 |
| GSE178557 | Testing | 8 | 35733354 |

"clusterProfiler" package in R. The GO analysis encompassed cell composition (CC), biological processes (BP), and molecular functions (MF). GO terms with a P-value less than 0.05 were deemed significantly enriched. For KEGG pathway enrichment, we used the "c2.cp.kegg.v7.0.symbols.gmt" gene set from the Molecular Signature Database (MSigDB; version 7.1), setting significance thresholds at a P-value < 0.05.

## Identification of signature genes associated with cellular senescence in RA using machine learning

The random forest algorithm was performed to identify the important cellular senescence-associated DEGs. This algorithm is a type of ensemble learning method that combines many decision trees to make a single decision based on the outputs of several classifiers together [14]. Each decision tree within the forest is constructed by selecting several samples from the original dataset using the bootstrapping method and is subsequently trained using a feature set selected by the randomly bagged mechanism [15]. Afterward, decisions made by numerous distinct individual trees are subjected to voting. The class prediction is subsequently assigned to the class with the highest number of votes obtained from the voting process [16]. In our study, we utilized the "randomForest" package for random forest analysis and the "glmnet" package for LASSO regression analysis to identify key cellular senescence-associated genes in RA. Based on these genes, we developed a predictive model, termed CSscore. LASSO, combining subgroup selection and linear regression, operates on the principles of ordinary least squares while constraining the sum of the absolute regression coefficients to a predetermined constant [17]. By implementing LASSO regression, certain regression coefficients are reduced to zero, resulting in only genes with non-zero regression coefficients being retained in the final model. Finally, LASSO regression was used to create the cellular senescence-related score (CSscore) model that depended on the hub cellular senescence related genes. The formula for the CSscore model is as follows:

$$\text{CSscore} = \sum\nolimits_{i \in S} \beta_i * \text{Exp}_i$$

Where S represents the set of cellular senescence-associated genes, $\beta_i$ is the regression coefficient of LASSO regression, $\text{Exp}_i$ is the expression level of gene i. Then the predictive ability of the CSscore model was assessed using receiver operator characteristic (ROC) curve via the "pROC" package.

## Difference of immune infiltration between RA and normal samples

Bioinformatic algorithms such as CIBERSORT, xCell, and single sample gene set enrichment analysis (ssGSEA) were employed to determine the immune cell infiltration in the samples. CIBERSORT is a de-convolution algorithm used to evaluate the proportion of immune cells in complex tissues from their gene expression profiles [18]. xCell is a new gene signature-based approach, inferring infiltration of immune and stromal cell subgroups via "xCell" R package [19]. ssGSEA algorithm was also utilized to infer the infiltration of immune cells using published immune signatures via "GSVA" package in R software, which revealed enrichment of diverse immune cell populations in patients [20]. The Wilcoxon rank sum test was used to evaluate the statistical significance of differential immune cell infiltration. The expression level between immune-related genes was evaluated using the non-parametric Spearman correlation test.

## Consensus clustering analysis

We conducted a consensus clustering analysis to classify RA samples based on cellular senescence-associated DEGs, using the "ConsensusClusterPlus" package. This analysis, conducted

over 1,000 iterations, aimed to ensure the robustness of our subgroup classification. Additionally, we utilized Spearman correlation analysis to further validate the classification's stability. The analysis parameters included a maximum of seven clusters, 50 replicates, a sample proportion of 0.8, and a feature-to-sample ratio of 0.8. We employed hierarchical clustering with Spearman's distance as the metric. Post clustering, we examined the expression of cellular senescence-related DEGs and immune cell infiltration across the identified subtypes.

## Validation of the predictive ability of the CSscore model in the external datasets

The predictive value of the CSscore model in four independent datasets [GSE12021, GSE55235, GSE77298, GSE178557] was validated using the ROC curve analysis. Details of the datasets are provided in Table 1.

## Association between signature genes and drug sensitivity

To further explore whether the signature genes could be used as potential drug targets, we first using the "pRRophetic" R package to estimate the sample's drug sensitivity IC50 value for 18 drugs from CCLE database. pRRophetic built a model based on the known cell line expression matrix and drug sensitivity information, then the new expression matrix is predicted. Subsequently, Spearman's rank correlation analysis was performed to assess the association between the expression of signature genes and drug sensitivity. Besides, we also searched the treatments for rheumatoid arthritis in Drugbank database.

AutoDock is an approach to drug design through the characterisation of receptors and the mode of interaction between receptors and drug molecules [21]. A theoretical simulation method that focuses on the study of intermolecular (e.g., ligand-receptor) interactions and predicts their binding modes and affinities. AutoDock is the process of simulating molecular recognition in a computer, with the aim of finding the optimal binding conformation of a protein and its ligand and ensuring that the overall binding free energy of the complex is minimised. To further explore whether the signature genes we identified could be potential targets for these retrieved rheumatoid arthritis drugs, AutoDock was performed. Detailedly, the crystal structures of signature genes were downloaded from RCSB Protein Data Bank (PDB, https://www.rcsb.org/) [22] and the 3D structure of small molecule compound was downloaded from the PubChem Compound database. The downloaded complex was embellished by PyMol2.3.0 to remove original water molecules and phosphates. Moreover, the AutoDock Tools 4.2.6 [21] (https://autodock.scripps.edu/) was used to prepare receptors, including adding Gasteiger charges, merging non-polar hydrogen bonds and setting docking parameters. A box of 60*60*60-point grid (0.375-Å spacing between the grid points) and the affinity maps were generated and processed using AutoGrid 4.2 by default setting. For each docking case, 200 Lamarckian genetic algorithm runs were processed by default setting using AutoDock 4.2.6. To further confirm the binding of small molecule compound to the selected genes, we have also obtained the structures of proteins encoded by the selected genes from AlphaFold and then re-performed AutoDock. The top-scored hit was chosen and visualized for further analysis and the PyMOL was used to preparse all the molecular graphics.

## Statistical analysis

For our statistical analyses, we employed R software, specifically version 3.6.1, which is available at [http://www.R-project.org]. To analyze the significance of differences between the identified subgroups in our study, we utilized the Wilcoxon rank sum test. We established our threshold for statistical significance at a p-value of less than 0.05.

## Results

### Differentially expressed cellular senescence related genes between RA and normal samples

In this study, we aimed to identify cellular senescence-related biomarker genes for predicting RA. A detailed flowchart of our study is shown in Fig 1. Differential expressed analysis was first carried out after pre-processing and normalisation of the gene expression in different RA cohorts. In total, there were 2050 DEGs identified using the filtering criteria of P-value less than 0.05 and the absolute value of log2 FC greater than 0.5 (Fig 2A). Next, the Venn diagram showed that 29 overlapping genes were identified between 2050 DEGs and 279 cellular senescence-related genes obtained from the CellAge database (Fig 2B). Among the 29 DEGs associated with cellular senescence, there were 19 genes that were up-regulated expressed and 10 genes were down-regulated expressed (S2 Table). Subsequently, the random forest algorithm and LASSO regression analysis were respectively performed. The results showed that there were 3 cellular senescence-related DEGs identified by random forest algorithm and LASSO regression analysis respectively (Fig 2C and 2D). Then, 3 out of the 29 cellular senescence-related DEGs ultimately remained as hub genes, including DHX9, CYR61 and ITGB (Fig 2C and 2D). A large number of pre-clinical research have shown that cysteine-rich 61 (CYR61) plays an important role in rheumatoid arthritis pathogenesis [23]. Zhu et al. demonstrated that the CYR61 is involved in the development of RA by promoting the production of pro-IL-1β by fibroblast-like synoviocytes via an AKT-dependent NF-κB signaling pathway [24]. DHX9 can also enhanced NF-kappaB-dependent IL-6 promoter activation, indicating that gene DHX9 may also associate with RA development [25].

### GO and KEGG function enrichment analysis

In order to understand the biological underlying mechanisms in which the DEGs associated cellular senescence are involved, we then carried out the GO and KEGG functional enrichment analysis in those genes. The GO functional analysis results showed that these genes were significantly enriched in the biological pathways involved in regulating D-binding transcription factors, regulation of MAP kinase activity, and JNK cascade. As for the cellular component, those genes were primarily enriched in the spindle microtubule, XY body, early phagosome, and npBAF complex. Besides, those genes were also significantly enriched in the molecular function of protein tyrosine kinase activity, histone binding, and integrin binding (Fig 3A and S3 Table). KEGG functional enrichment analysis revealed that the DEGs associated with cellular senescence were significantly enriched in p53 signaling pathway, ErbB signaling pathway, Gap junction, and osteoclast differentiation (Fig 3B and S4 Table). Osteoclasts play an important function in the maintaining, repairing, and remodeling bones in the adult vertebral skeleton through their bone resorbing capacity. And RA was proven to be related to increased osteoclast activity [26–28].

### Construction of the diagnosis model for RA patients

To assess the diagnostic value of the three signature genes identified from the previous processes, ROC curves analysis based on the expression of these three genes were carried out. The results demonstrated that these genes had good predictive ability in the training datasets with AUCs ranging from 0.848 to 0.961 (S1 Fig), indicating good diagnostic performance of these genes in the training cohort. Among them, DHX9 had the highest predictive ability in RA samples (AUC = 0.961), followed by ITGB4 (AUC = 0.939), and CYR61 (AUC = 0.848). To further explore the diagnostic value using these three genes jointly, a CSscore model was built via the

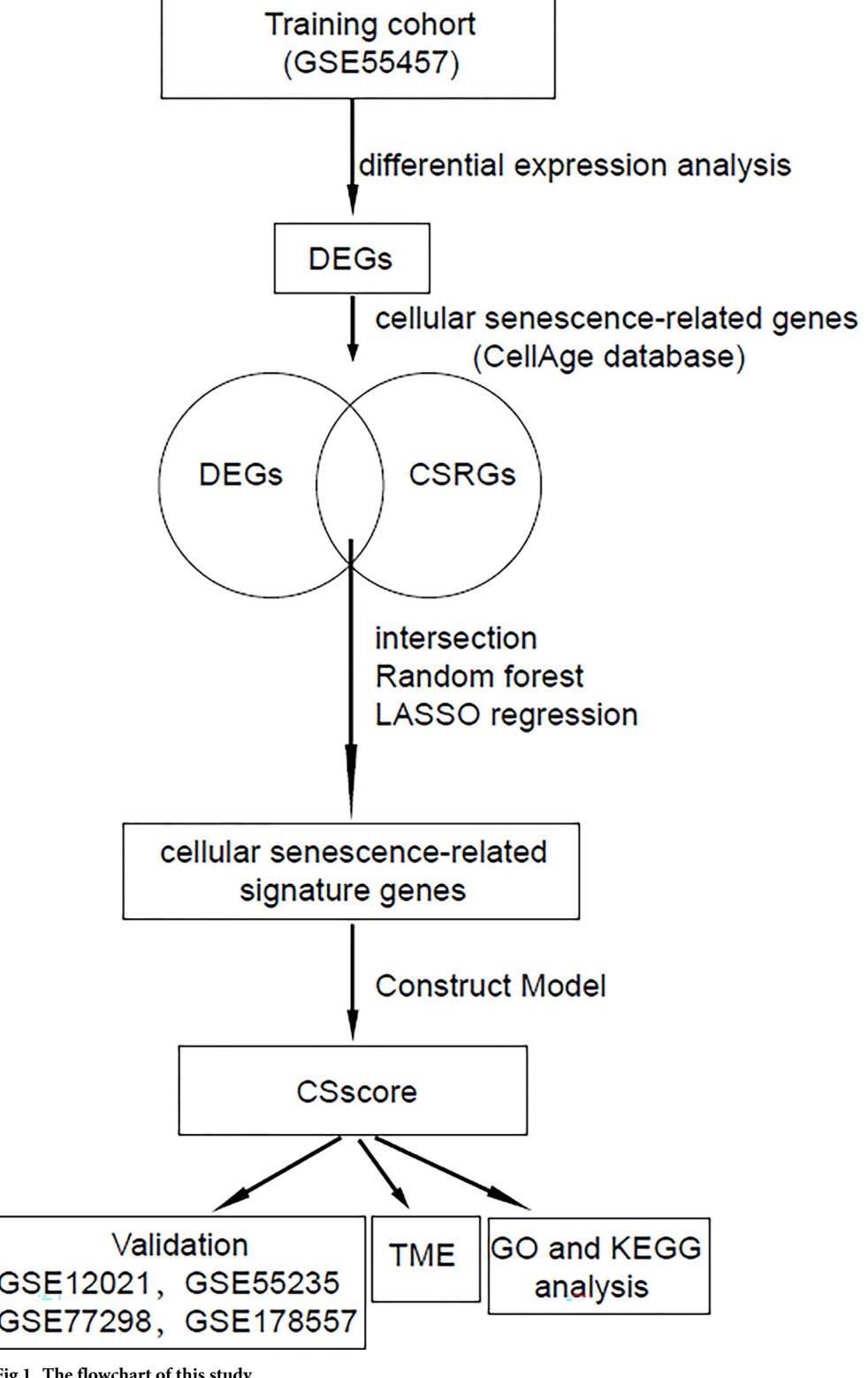

**Fig 1. The flowchart of this study.**

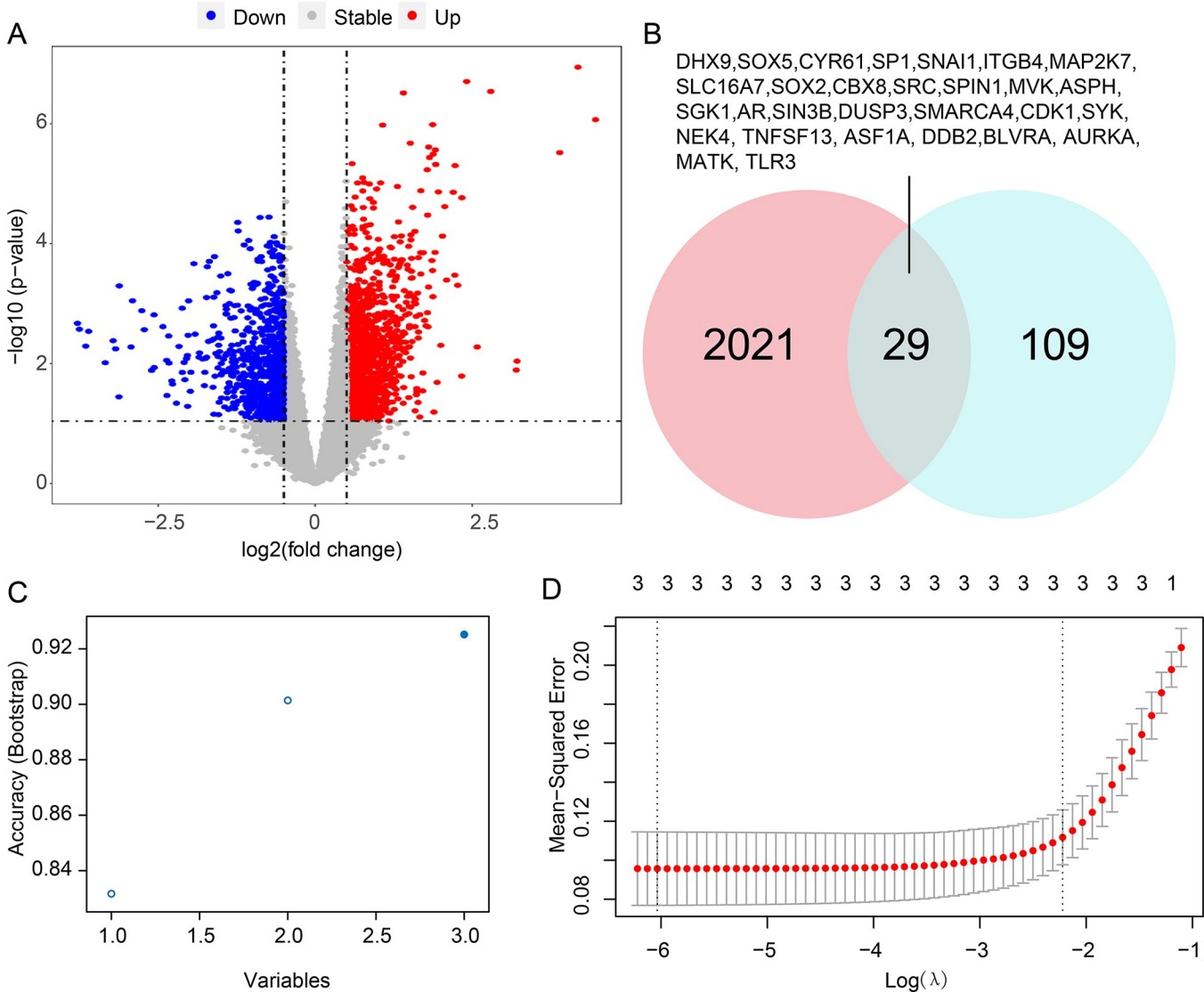

**Fig 2. Screening of the cellular senescence-associated signature genes. (A)** The Volcano plots showing significantly DEGs (P-value < 0.05, |log2 Fold Change| > 0.5) in RA vs. normal samples. **(B)** The overlapping genes between DEGs and cellular senescence-associated genes. **(C)** The dot plots show the accuracy of model in each time. **(D)** Partial likelihood deviance for the LASSO regression.

LASSO regression. CSscore = -0.08546212 * DHX9–0.08646213 * CYR61–0.11403349 * ITGB + 2.78665503. The ROC curve analysis indicated that this model exhibited a higher predictive ability for RA than the single gene model (AUC = 0.987, Fig 4).

Moreover, the predictive ability of the CSscore model for RA samples was validated in the four independent external cohorts. In GSE12021 cohort, which consist of 57 RA samples, the AUC of the CSscore model was 0.932 (Fig 5A). In GSE55235 cohort, the AUC of our CSscore model reached 1, indicating that the model could completely predict RA patients accurately (Fig 5B). Because GSE55235 cohort is affiliated with the GSE55457 cohort (training cohort), which results in such high predictive power. As for the GSE77298 and GSE178557 cohorts, the AUC of CSscore model for RA were both greater than 0.8 (Fig 5C and 5D). These findings showed that the CSscore model exhibited excellent diagnostic value for RA in both training and testing datasets, indicating the CSscore model could serve as potential diagnostic biomarker for RA.

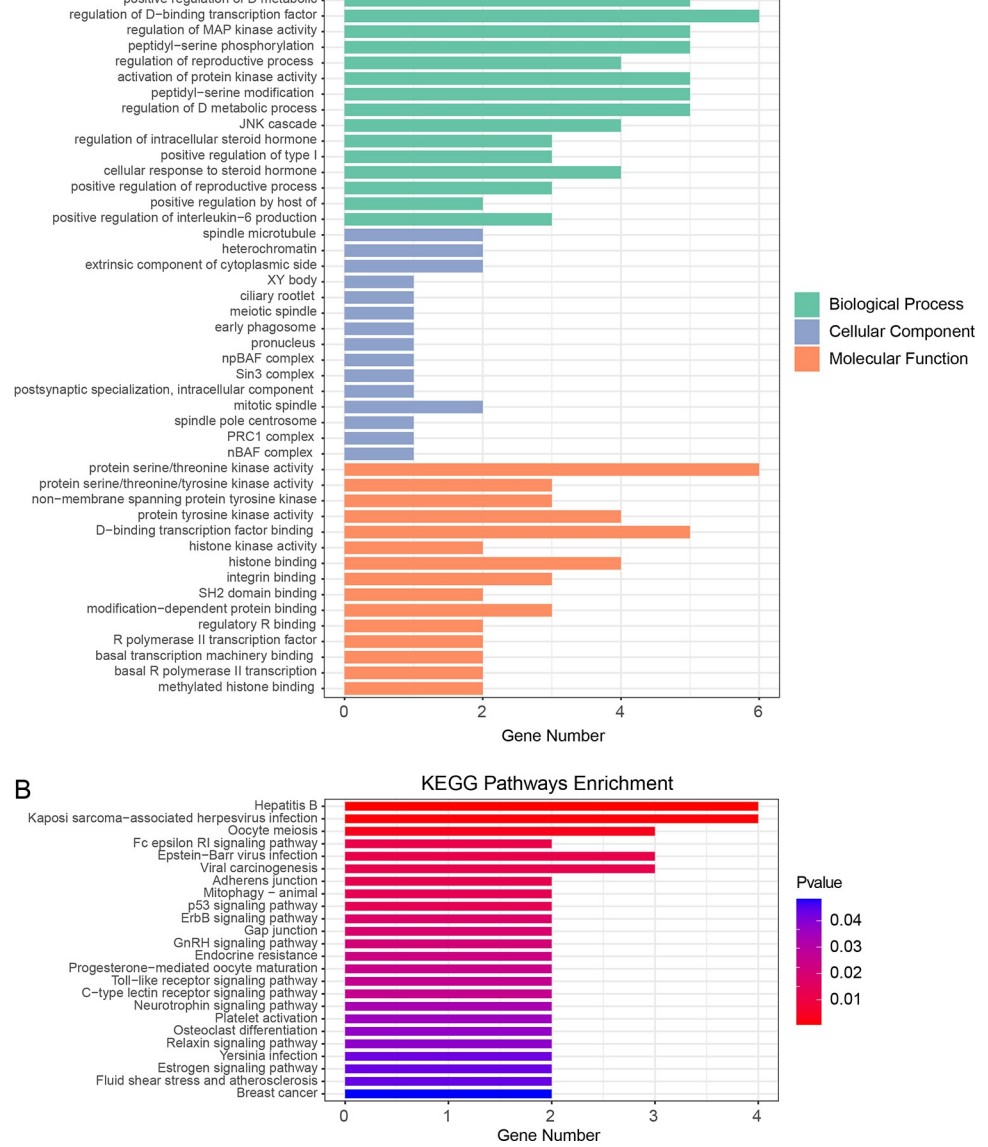

**Fig 3. Function enrichment analysis. (A)** GO functional analysis of the cellular senescence related DEGs between RA and control. **(B)** KEGG pathway analysis of the cellular senescence related DEGs between RA and control.

## Difference in immune infiltration between RA and normal samples

RA is a chronically inflamed disease with an important autoimmune element. Both innate and adaptive mechanisms interact closely to enhance the inflammation of joint if not treated appropriately. We then investigate the differences in the infiltration of immune cells between RA and normal samples. The infiltration of immune cells for samples was estimated using the bioinformatic methods, which included CIBERSORT [18], xCell [19], and ssGSEA [20]. As shown in the CIBERSORT analysis, there are 4 out of 22 immune cells significantly different infiltrated between RA and normal samples, including Macrophages M1, Mast cells activated, T cell CD4 memory resting, and T cells follicular helper (Fig 6A). As for the ssGSEA, there are 10 out 24 immune cells significantly different infiltration, especially including aDC, B cell, T

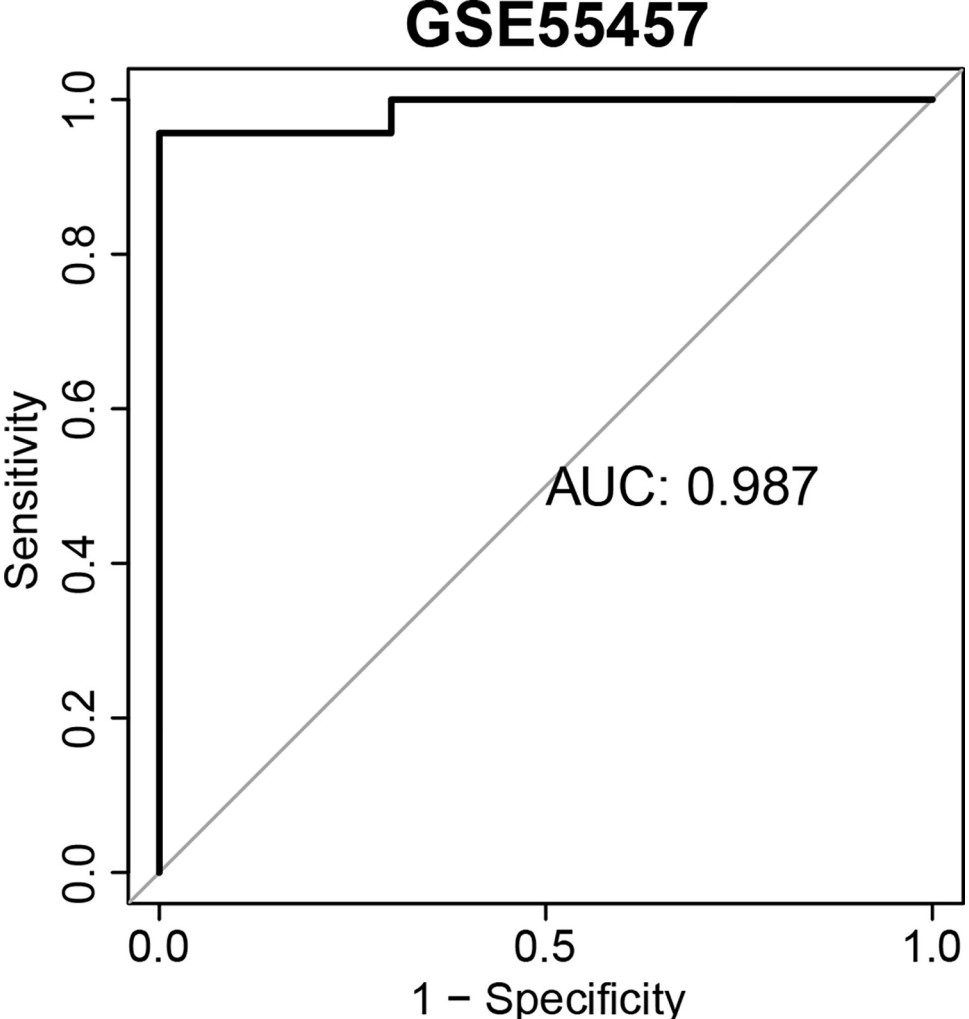

**Fig 4. The ROC curve analysis to assess the predictive ability of the CSscore model in training dataset (GSE55457).**

cell, and Th2 cells (Fig 6B). Furthermore, there are 11 immune cells significantly different infiltrated between RA and normal samples (Fig 6C), of which CD8+ Tcm were the most significantly different. In each of the three different immunoinfiltration algorithms, a significantly distinct T cell or B cell subtype was infiltrated, indicating that the immune cells are important in RA pathogenesis. Increasing studies proved that the T cell or B cell senescence was a critical driver in RA [5,7,29,30], opening new possibilities for immune-modulating therapy by recovering the functional integrity of senescent T/B cells.

Moreover, we explored the correlations between immune checkpoint genes through Spearman correlation analysis. A significant positive correlation between CD40 and CD86 (r = 0.716, P < 0.05) was found, and a significant negative correlation between CD40 and NT5E was found (r = -0.610, P < 0.05) (Fig 7). Iscalimab is a prospective CD40-CD154 co-stimulation pathway blocker with potential applications in transplantation and other autoimmune-related diseases [31]. Wan et al. found that CD86 and CD40 were down-regulated in mesenteric lymph nodes and that CD40, CD40L, CD28, CD80, and CD86 were also down-

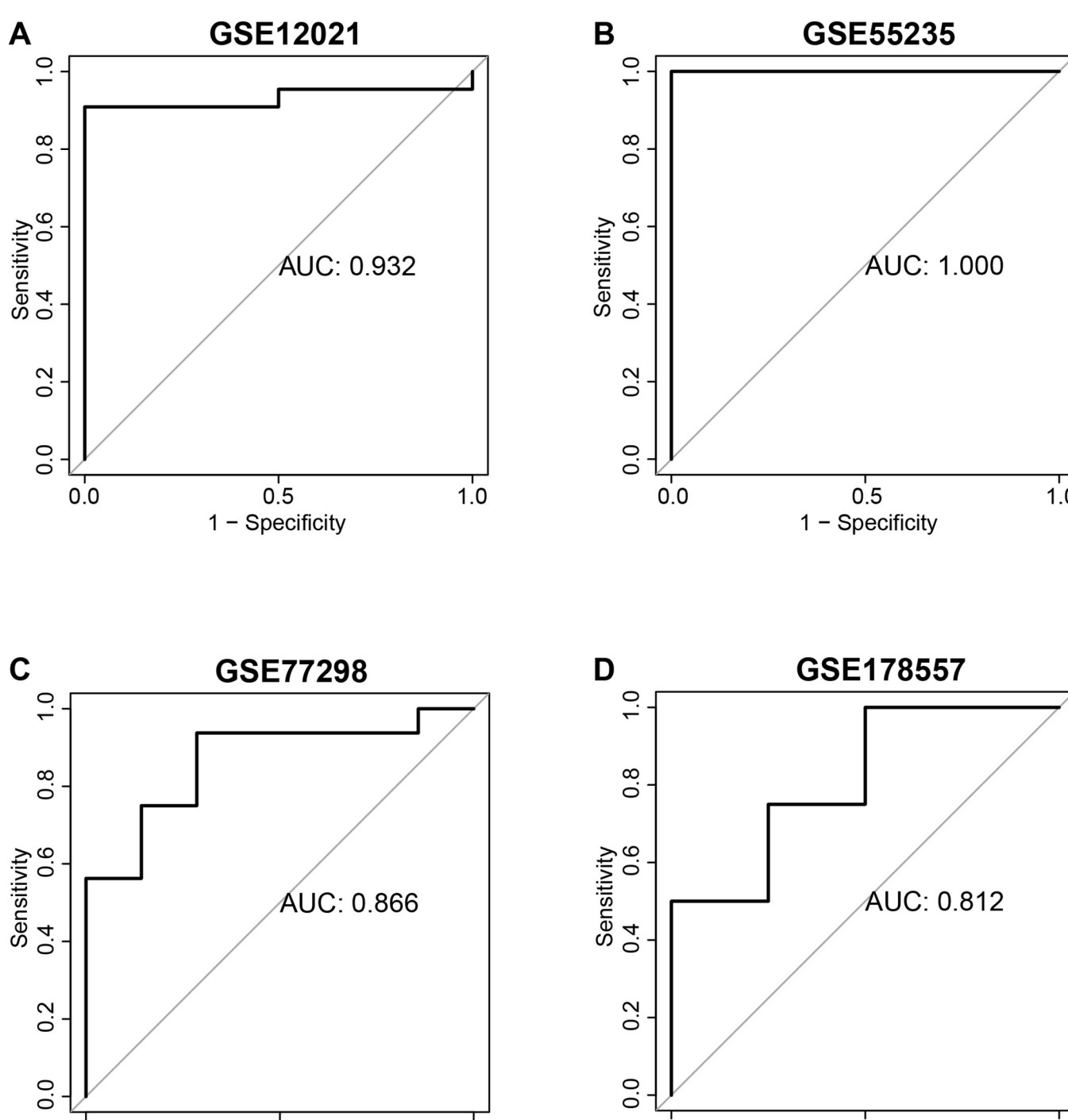

**Fig 5. Validate the predictive value of the SCscore model in four independent external cohorts.** (**A**) The ROC curve of CSscore model in GSE12021. (**B**) The ROC curve of CSscore model in GSE55235. (**C**) The ROC curve of CSscore model in GSE77298. (**D**) The ROC curve of CSscore model in GSE178557.

regulated in the rat ankle upon stimulation of immune checkpoint molecules [32]. That is to say, the expressions of CD40 and CD86 are coordinate regulation.

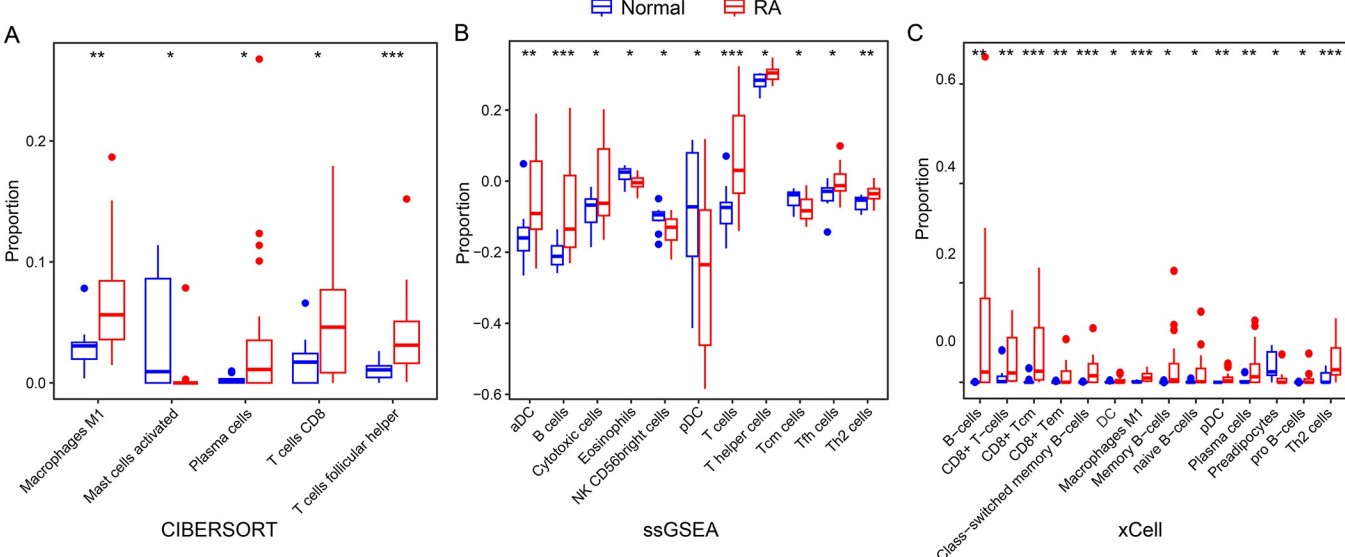

**Fig 6. Compare the infiltration of immune cells between RA and normal samples.** The infiltration of immune cells was estimated by (**A**) CIBERSORT (**B**) ssGSEA (**C**) xCell.

## Immune infiltration landscape of distinct subtypes

The consensus clustering analysis was performed using the "ConsensusClusterPlus" package based on the expression of DEGs associated with cellular senescence. To ensure the robustness of the classification, the consensus clustering analysis was performed for 1,000 repetitions and the Spearman correlation analysis was used. Through consensus clustering analysis, RA samples were grouped into two subgroups on the basis of the 29 cellular senescence-related DEGs mRNA expression, namely, subtype C1 (n = 12), subtype C2 (n = 21) (Fig 8A). Our results showed that BLVRA, ASF1A, MATK, and TLR3 tended to be under-expressed in subtype C1 samples, while high-expressed in subtype C2 samples (Fig 8B). We also found that genes SMARCA4, SIN3B, and MA2K7 were highly expressed in subtype C1 samples, while under-expressed in subtype C2 samples (Fig 8B). To elucidate the differences in the immune infiltration landscape among three subgroups, the CIBERSORT algorithm was used to infer immune cell infiltration and evaluate HLA and immune checkpoint gene sets expression. We also revealed significant differential expressions in 19 immune checkpoint genes, including CD160, CD27, CD48, CD96, TNFRSF14, and TNFRSF17 (Fig 8C). Notably, these genes were significantly high expressed in the subtype C2 samples, except for B2M, HLA-A, HLA-C, IL6R, PDCD1, and PVR. And we also found that mast cells activated and monocytes in subtype C1 were remarkably increased versus subtype C2 (P < 0.05) (Fig 8D). Compared with subtype C1, the infiltrating of Macrophages M1, T cells follicular helper, and T cells gamma delta were significantly increased in subtype C2 samples. These findings demonstrated that the immune infiltration landscape in RA samples is differentially affected by distinct expression patterns of DEGs associated with cellular senescence.

## Association between signature genes and drug sensitivity

To further explore whether the signature genes could be used as potential drug targets, we first using the "pRRophetic" R package to estimate the sample's drug sensitivity IC50 value for 18 drugs from CCLE database. Subsequently, Spearman's rank correlation analysis was

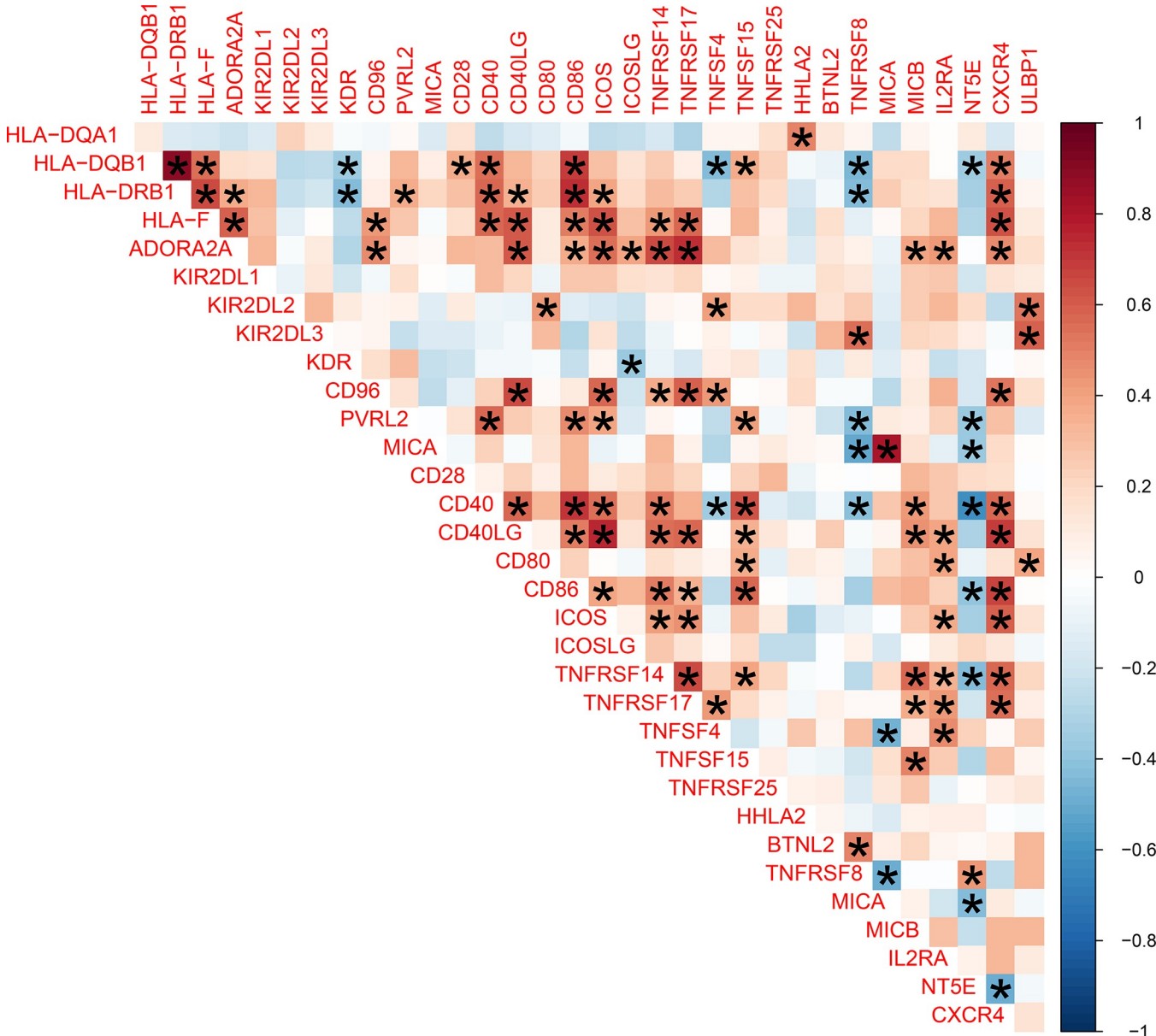

**Fig 7. Correlation analysis between the expression level of immune checkpoint related genes.**

performed to assess the association between the expression of signature genes and drug sensitivity. The results showed that the expression of DHX9 was significantly negative correlation with the sensitivity of PLX4720, PD.0325901, Axitinib, and Bosutinib (Fig 9A). The expression of CYR61 was significantly negative correlation with the sensitivity of Axitinib, and positive correlation with the sensitivity of PF2341066, PD.0332991, Cisplatin (Fig 9A). The expression of ITGB4 was significantly negative correlation with the sensitivity of AZD.0530, and Dasatinib (Fig 9A). We first searched the Drugbank database for RA treatment drugs, ibuprofen and indometacin are available for RA treatment. The results of AutoDock displayed that the binding energy of gene CYR61 and ibuprofen is -6.03 kcal/mol (Fig 9B), while the binding energy of gene CYR61 and indometacin is -6.9 kcal/mol (Fig 9C). Moreover, the binding energy of

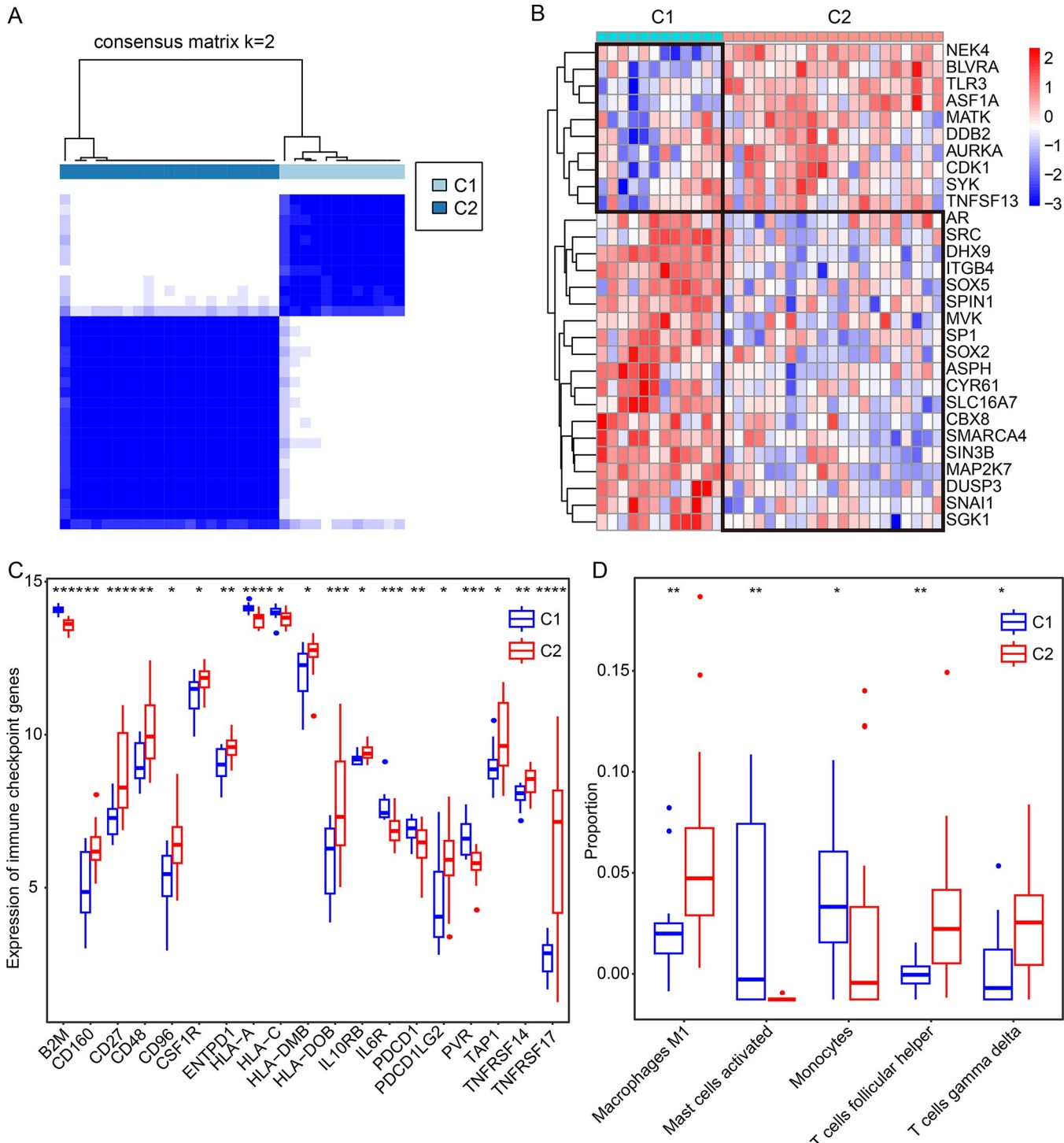

**Fig 8. Consensus clustering analysis. (A)** The samples were clustered into three subtypes based on the expression of cellular senescence-related DEGs by consensus clustering method. **(B)** The expression heatmap of cellular senescence-related DEGs between three subtypes samples. **(C)** The expression boxplot of immune checkpoint related genes between three subtypes samples. **(D)** The infiltration boxplot of immune cells between three subtypes samples.

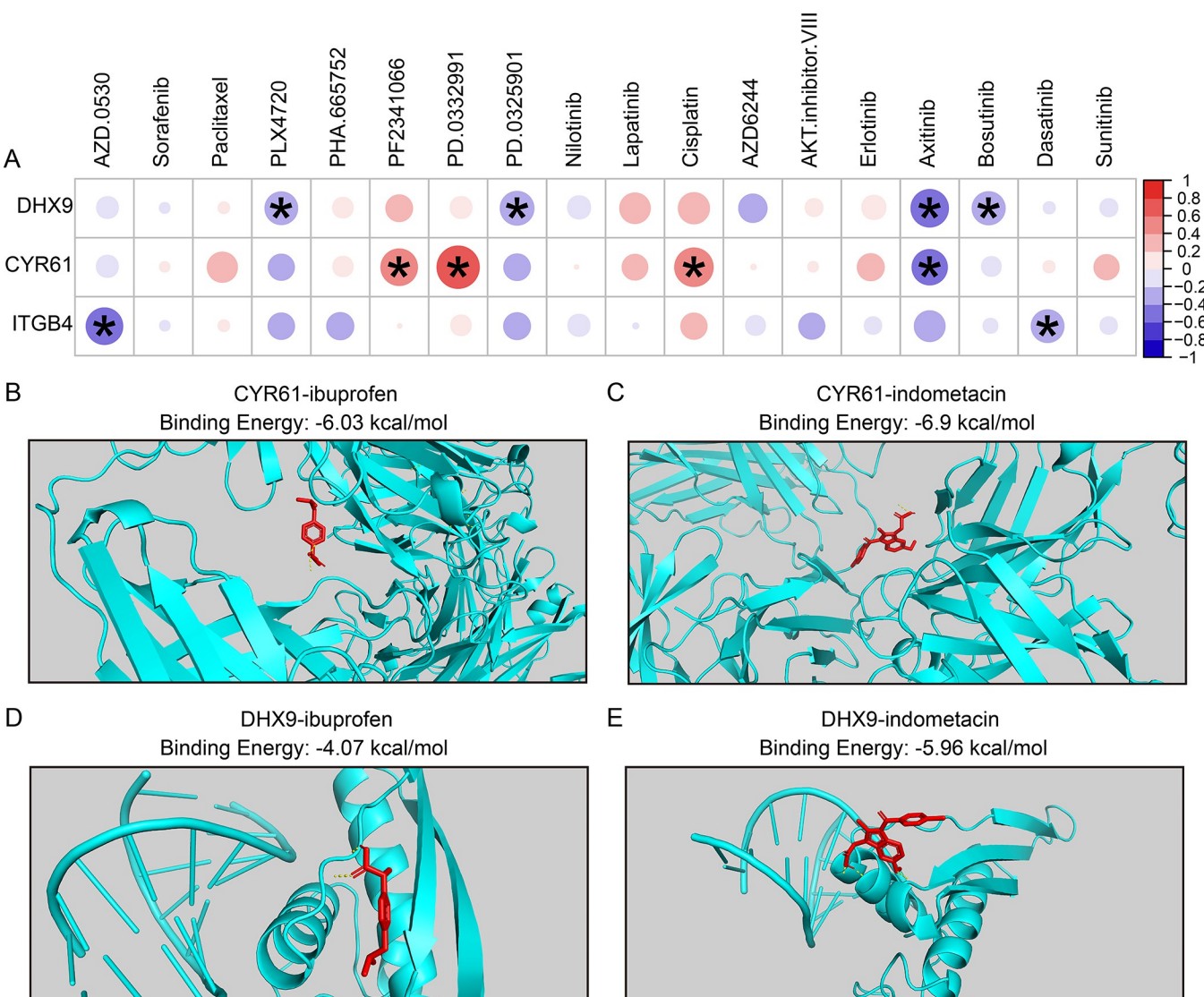

**Fig 9. Association between signature genes and drug sensitivity. (A)** Correlation analysis between the expression level of signature genes and the IC50 values of drugs. **(B)** The AutoDock results of gene CYR61 and drug ibuprofen. **(C)** The AutoDock results of gene CYR 61 and drug indometacin. **(D)** The AutoDock results of gene DHX9 and drug ibuprofen. **(E)** The AutoDock results of gene DHX9 and drug indometacin.

gene DHX9 and ibuprofen is -4.07 kcal/mol (Fig 9D), while the binding energy of gene DHX9 and indometacin is -5.96 kcal/mol (Fig 9E). To further confirm the binding of small molecule compounds to the selected genes, we have also obtained the structures of proteins encoded by the selected genes from AlphaFold and then re-performed AutoDock. As shown by the docking results, ibuprofen and indometacin both possessed high binding affinity with an average of -7.25 kcal/mol. Specifically, the binding energy of gene CYR61 and ibuprofen is -6.55 kcal/mol (S2A Fig), while the binding energy of gene CYR61 and indometacin is -9.01 kcal/mol (S2B Fig). Moreover, the binding energy of gene DHX9 and ibuprofen is -6.43 kcal/mol (S2C Fig), while the binding energy of gene DHX9 and indometacin is -7.02 kcal/mol (S2D Fig). Molecular docking (AutoDock) provides a computationally efficient approach to evaluate molecular interactions; however, it may inadequately capture certain specific interactions, such as hydrogen bonding. Molecular dynamics simulations offer a more comprehensive framework for

understanding binding affinities and mechanisms. In our future studies, we could leverage these simulations to complement and refine the insights gained from docking analyses. These findings suggested that the signature genes (CYR61 and DHX9) could be used as potential drug targets for RA treatment.

## Discussion

RA is a prevalent autoimmune disease characterized by synovitis and hyper-plasia ("swelling"), production of autoantibodies (rheumatoid factor and anti–citrullinated peptide antibodies [ACPA]), destruction of cartilage and bone ("deformity"), and systemic features such as cardiovascular, pulmonary, psychological, and skeletal abnormalities [33]. While the RA pathogenesis is still poorly understood, the last decade has provided us closer to understanding the cellular and molecular mechanisms involved. Researchers found that numerous aging-related diseases, such as tumor, cardiovascular disease, RA, are associated with a decrease in immune competency accompanied by an increase in pro-inflammatory. As a result, the process of immune senescence is situated at the core of aging biology [34,35]. For instance, Oscar et al. verified that the decrease in numbers of senescent immune cells, including CD8+ T-cells, leukocytes and lymphocytes in women diagnosed with breast cancer may contribute to cancer development [36]. Another randomized controlled study also proved that peripheral T-cells linked to immunosenescence could take involvement in cancer occurrence [37]. Furthermore, aging causes immune system dysfunction and leads to various aging-related diseases, which is called immunosenescence [38]. Previous literature also indicated that immunosenescence leads to a significant decrease in protective immune response and an accompanying rise in the inflammatory responses of tissues [39]. In addition, early signs of senescence have been linked to RA. And the concept of immunosenescence in the developing and progressing of RA has garnered significant attention. However, the molecular mechanisms are still poorly understood and there has not yet been a bioinformatics analysis of aging-related genes in RA.

Our study provides novel insights into the pathogenesis of rheumatoid arthritis (RA) by integrating bioinformatics and machine learning. We extracted RA datasets and cellular senescence-related genes from the GEO and CellAge databases, respectively. Our analysis identified 29 differentially expressed genes (DEGs) associated with cellular senescence. Using GO and KEGG analyses, we explored the biological mechanisms of these DEGs, suggesting their role in RA development. Key cellular senescence-related markers, DHX9, CYR61, and ITGB, were identified using random forest and LASSO Cox regression methods. Particularly, CYR61, known for its role in cell proliferation and migration, emerged as a significant factor in RA pathogenesis based on extensive pre-clinical studies. Our approach underscores the potential of using cellular senescence markers for diagnosing and understanding RA [40]. Zhu et al. showed that the CYR61 is involved in RA development by enhancing the production of proIL-1β, which may exhibits powerful proinflammatory activity in the RA development [24]; Zhai et al. demonstrated that CYR61 induce the production of MMP-3 production in RA FLS through the P38, JNK-dependent AP-1 signaling pathway [41]. And previous studies have found that the protease MMP-3 is considered to be responsible for joint destruction, with higher serum MMP-3 levels in RA patients [42,43]; Lin et al. uncovered that Cyr61 promotes Th17 differentiation in RA by inducing IL-6 production [44]. Additionally, the differentiation and plasticity of Th17 cells is closely linked to RA disease activity both in vivo and in vitro [45]. As for DHX9, it belongs to DExD/H helicase family and is involved in detecting both short and long poly I:C in myeloid dendritic cells (mDCs) and in mediating antiviral innate immune response [46,47]. Studies have suggested that Dhx9 can modulate CD8+ T cell–mediated antiviral response by preventing T cells from dying and modulating genes essential for the

differentiation of effector T cells [48]. Specifically, recent studies observed that aging causes a reduction in the dynamical range of the anti-viral innate immune response [49]. Aging is associated with impaired immune response to viral pathogens which can further affect the production of soluble pattern recognition molecules and inflammasome. Indeed, reduced TLRs expression in the older individuals is associated with with reduced inflammatory cytokine responses and impaired pattern recognition [50]. From the aforementioned analysis, we deduced that aging accompanies T cell senescence and a progressive decline in immune response and function, leading to greater vulnerability to viral infection.

We acknowledge that minimizing the number of selected genes can potentially reduce costs and improve interpretability. However, this reduction might also negatively impact sensitivity and specificity, possibly increasing the risk of false negatives. Future studies could explore this trade-off in greater detail, performing comprehensive analyses to evaluate sensitivity and specificity as a function of gene selection thresholds. Furthermore, while we chose an RF-based strategy for its robustness in this context, exploring XGBoost or other advanced machine learning architectures in future studies could provide additional insights into the potential contribution of a larger gene set.

In our research, individual gene models demonstrated that the three identified genes had strong diagnostic capabilities for rheumatoid arthritis (RA). We constructed an integrated diagnostic model (CSscore) using these signature genes, which showed superior predictive accuracy compared to single gene models. Our analysis also revealed distinct differences in immune cell infiltration and immune checkpoint gene expression between RA and normal samples, particularly noting an enrichment of B and T cell subtypes in RA samples. The predictive value of our method was further confirmed through external validation cohorts, exhibiting excellent predictive abilities. Additionally, we classified RA samples into three unique subgroups based on cellular senescence-associated DEGs and observed distinct effects of these DEGs on immune cell infiltration. This study enhances understanding of the molecular mechanisms of cellular senescence in RA, potentially offering new directions for diagnosis and treatment.

## Supporting information

**S1 Fig. The diagnostic value of cellular senescence related signature genes.**
(TIF)

**S2 Fig. Molecular docking analysis.**
(TIF)

**S1 Table. The cellular senescence-related genes from CellAge database.**
(DOCX)

**S2 Table. The cellular senescence related DEGs between RA and normal samples.**
(DOCX)

**S3 Table. The GO terms that the cellular senescence related DEGs involved in.**
(DOCX)

**S4 Table. The KEGG pathways that the cellular senescence related DEGs involved in.**
(DOCX)

## Author Contributions

**Conceptualization:** Tianhua Yu.

**Data curation:** Tianhua Yu, Zhichao Wang.

**Formal analysis:** Zhichao Wang.

**Methodology:** You Ao.

**Software:** Qing Lan, Zhichao Wang.

**Supervision:** You Ao.

**Visualization:** Qing Lan, Jing Zhang.

**Writing – original draft:** You Ao.

**Writing – review & editing:** You Ao, Jing Zhang.

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
