## [Decision Letter · Decision Letter 0]

10 Mar 2024

PONE-D-24-02627Cellular Senescence-Associated Genes in Rheumatoid Arthritis: Identification and Functional AnalysisPLOS ONE

Dear Dr. Zhang,

Thank you for submitting your manuscript to PLOS ONE. After careful consideration, we feel that it has merit but does not fully meet PLOS ONE’s publication criteria as it currently stands. Therefore, we invite you to submit a revised version of the manuscript that addresses the points raised during the review process.

We look forward to receiving your revised manuscript.

Kind regards,

Gurudeeban Selvaraj

Academic Editor

PLOS ONE

Journal Requirements:

Reviewers' comments:

Reviewer's Responses to Questions

**Comments to the Author**

1. Is the manuscript technically sound, and do the data support the conclusions?

Reviewer #1: No

2. Has the statistical analysis been performed appropriately and rigorously? 

Reviewer #1: No

3. Have the authors made all data underlying the findings in their manuscript fully available?

Reviewer #1: No

4. Is the manuscript presented in an intelligible fashion and written in standard English?

Reviewer #1: No

5. Review Comments to the Author

Reviewer #1: The authors obtained the RA dataset from the GEO database and the genes associated with cellular senescence from the CellAge database. The authors' first step was to identify degs associated with cellular senescence. then, the authors used GO and KEGG analyses to delve deeper into the biological processes of these senescence-associated degs. Subsequently, the authors used Random Forest analysis to further identify significant deg associated with senescence. in addition, the authors used LASSO Cox regression analysis to select key genes associated with cellular senescence in RA, which led to the development of the CSscore prediction model. The authors further confirmed the validity of our approach through validation against an external cohort. Ultimately, the authors divided the RA samples into three different groups based on senescence-related gender differences and performed a comparative analysis of their immune landscapes

1. In the differential expression analysis, the points (genes) shown in the volcano diagram are so few that it is certain that the dataset used contains the whole transcriptome?

2. The authors further screened the genes by RF algorithm and obtained the core genes for model construction by taking the intersection set. However, these steps are all lacking in the image.

3. The authors validated by ROC curve in GSE55235 and got the result that AUC is equal to 1. The authors should know about overfitting.

4. the authors validated using an external dataset and should not only validate the predictive power, but also the expression difference.

5. when the authors performed the TME analysis, it seems that most of the immune cells are not statistically different in different subgroups.

6. Correlation analysis in red indicates a negative correlation and in blue a positive correlation. This also seems to be the opposite of the conventional drawing method.

7. the clustering results are also poor.

8. the study also lacks validation from basic experiments.

Therefore, the manuscript is preliminary and crude and does not merit publication.

6. PLOS authors have the option to publish the peer review history of their article (what does this mean?). If published, this will include your full peer review and any attached files.

Reviewer #1: No

---

## [Author Response · Author response to Decision Letter 0]

29 Apr 2024

Dear Prof. Gurudeeban Selvaraj and reviewers:

On behalf of my co-authors, we thank you very much for giving us an opportunity to revise our manuscript, we appreciate editor and reviewers very much for their positive and constructive comments and suggestions on our manuscript entitled “Cellular Senescence-Associated Genes in Rheumatoid Arthritis: Identification and Functional Analysis” (PONE-D-24-02627). Those comments are all valuable and helpful for improving our paper. We have studied reviewer’s comment carefully and have made revision which marked in red in the paper. And we have tried our best to revise our manuscript according to the comments. The main corrections in the paper and the response to the reviewers’ comments are as follows:

Reply to Reviewer #1

Comments to the Author

The authors obtained the RA dataset from the GEO database and the genes associated with cellular senescence from the CellAge database. The authors' first step was to identify degs associated with cellular senescence. then, the authors used GO and KEGG analyses to delve deeper into the biological processes of these senescence-associated degs. Subsequently, the authors used Random Forest analysis to further identify significant deg associated with senescence. in addition, the authors used LASSO Cox regression analysis to select key genes associated with cellular senescence in RA, which led to the development of the CSscore prediction model. The authors further confirmed the validity of our approach through validation against an external cohort. Ultimately, the authors divided the RA samples into three different groups based on senescence-related gender differences and performed a comparative analysis of their immune landscapes.

REPLY: Thank you very much for your comments. Those comments are very constructive and helpful for improving our paper.

1. In the differential expression analysis, the points (genes) shown in the volcano diagram are so few that it is certain that the dataset used contains the whole transcriptome?

REPLY: Thank you for your comments. We are very sorry for our negligence of mistakes. In the differential expression analysis, the points shown in the volcano diagram is senescence-related genes but not all genes from whole transcriptome, thus resulting in very few points in the volcano diagram. According to your suggestions, we have revised the manuscript and Figure 2A as follows:

Page 8, section “Differentially Expressed Cellular Senescence Related Genes Between RA and Normal samples” in RESULTS

“In total, there were 2050 DEGs identified using the filtering criteria of P-value less than 0.05 and the absolute value of log2 FC greater than 0.5 (Figure 2A).”

“Next, the Venn diagram showed that 29 overlapping genes were identified between 2050 DEGs and 279 cellular senescence-related genes obtained from the CellAge database (Figure 2B).”

Page 16, section “Figure legends”

“Figure 2. Screening of the cellular senescence-associated signature genes. (A) The Volcano plots showing significantly DEGs (P-value < 0.05, |log2 Fold Change| > 0.5) in RA vs. normal samples. (B) The overlapping genes between DEGs and cellular senescence-associated genes. (C) The dot plots showing the accuracy of model in each time. (D) Partial likelihood deviance for the LASSO regression.”

2. The authors further screened the genes by RF algorithm and obtained the core genes for model construction by taking the intersection set. However, these steps are all lacking in the image.

REPLY: Thank you for your comments. We are very sorry for our negligence of mistakes. In our study, there were 2050 DEGs identified using the filtering criteria of P-value less than 0.05 and the absolute value of log2 FC greater than 0.5. To further explore the differential expression pattern of senescence-related genes, we have added a Venn diagram to show the overlapping genes between 2050 and cellular senescence-related genes obtained from the CellAge database. Subsequently, the random forest algorithm and LASSO regression analysis were respectively performed. The results showed that there were 3 cellular senescence-related DEGs identified by random forest algorithm and LASSO regression analysis respectively. Then, 3 out of the 29 cellular senescence-related DEGs were ultimately remained as hub genes, including DHX9, CYR61 and ITGB.

According to your comments, we have added Figure 2C and Figure 2D to display the results. Besides, we have revised the manuscript as follows:

Page 8, section “Differentially Expressed Cellular Senescence Related Genes Between RA and Normal samples” in RESULTS

“Subsequently, the random forest algorithm and LASSO regression analysis were respectively performed. The results showed that there were 3 cellular senescence-related DEGs identified by random forest algorithm and LASSO regression analysis respectively (Figure 2C-D). Then, 3 out of the 29 cellular senescence-related DEGs were ultimately remained as hub genes, including DHX9, CYR61 and ITGB (Figure 2C-D).”

Page 16, section “Figure legends”

“Figure 2. Screening of the cellular senescence-associated signature genes. (A) The Volcano plots showing significantly DEGs (P-value < 0.05, |log2 Fold Change| > 0.5) in RA vs. normal samples. (B) The overlapping genes between DEGs and cellular senescence-associated genes. (C) The dot plots showing the accuracy of model in each time. (D) Partial likelihood deviance for the LASSO regression.”

3. The authors validated by ROC curve in GSE55235 and got the result that AUC is equal to 1. The authors should know about overfitting.

REPLY: Thank you for your comments. In GSE55235 cohort, the AUC of our CSscore model reached 1, indicating that our CSscore model is overfitting. But the GSE55235 cohort is used as validation cohort but training cohort, the AUC of our CSscore model reaching 1 can only indicate that the model could completely predict RA patients accurately. Besides, we also found that the GSE55235 cohort is affiliated with the GSE55457 cohort (training cohort), so we hold the opinion that maybe it is not overfitting.

 According to your comments, we have revised the manuscript as follows:

Page 10, section “Construction of the diagnosis model for RA patients”

“In GSE55235 cohort, the AUC of our CSscore model reached 1, indicating that the model could completely predict RA patients accurately (Figure 5B). Because GSE55235 cohort is affiliated with the GSE55457 cohort (training cohort), which results in such high predictive power. As for the GSE77298 and GSE178557 cohorts, the AUC of CSscore model for RA were both greater than 0.8 (Figure 5C-D). ”

4. the authors validated using an external dataset and should not only validate the predictive power, but also the expression difference.

REPLY: Thank you for your comments. According to your suggestions, we have added analysis to explore the expression values of signature genes in validation cohorts. Since only the gene ITGB4 was expressed in the validation set GSE12021 and the expression of CYR61, DHX9 was missing, we therefore validated the expression patterns of the signature genes only in the other three validation sets (GSE55235, GSE77298, GSE178557). The results showed that the expression levels of signature genes only in GSE55235 cohort were significantly different between the normal and RA sample groups, while in the remaining two validation datasets the differences were not significant (GSE77298, GSE178557) (Supplementary Figure S2). Although the difference in the expression of the signature genes in the GSE77298 and GSE178557 was non-significant, their expression still predicted patient classification with high predictive performance, suggesting that there is also heterogeneity in the expression of signature genes.

5. when the authors performed the TME analysis, it seems that most of the immune cells are not statistically different in different subgroups.

REPLY: Thank you for your comments. Indeed, most immune cells showed no statistical difference between RA and normal samples. But there still some immune cells were infiltrated differently in RA and normal samples. For example, in CIBERSORT analysis, there are 4 out of 22 immune cells significantly different infiltrated between RA and normal samples, including Macrophages M1, Mast cells activated, T cell CD4 memory resting, and T cells follicular helper (Figure 6A). In ssGSEA analysis, there are 10 out 24 immune cells significantly different infiltration, especially including aDC, B cell, T cell, and Th2 cells (Figure 6B). Furthermore, in xCell analysis, there are 11 immune cells significantly different infiltrated between RA and normal samples (Figure 6C), of which CD8+ Tcm were the most significantly different. According to your comments, to make our results more clear, we then show only the infiltration levels of differentially infiltrated immune cells in each algorithm. We have revised the Figure 6.

6. Correlation analysis in red indicates a negative correlation and in blue a positive correlation. This also seems to be the opposite of the conventional drawing method.

REPLY: Thank you for your comments. We are very sorry for our negligence of mistakes. According to your comments, we have modified the Figure 7 to show negative correlation in blue and positive correlation in red.

7. the clustering results are also poor.

REPLY: Thank you for your comments. According to your suggestions, we re-performed the cluster analysis to cluster samples using consensus cluster algorithm based on the expression of senescence-related DEG. Through consensus clustering analysis, RA samples were grouped into two subgroups on the basis of the 29 cellular senescence-related DEGs mRNA expression, namely, subtype C1 (n = 12), subtype C2 (n = 21)(Figure 8A). Our results showed that BLVRA, ASF1A, MATK, and TLR3 tended to be under-expressed in subtype C1 samples, while high-expressed in subtype C2 (Figure 8B). We also found that genes SMARCA4, SIN3B, and MA2K7 were high-expressed in subtype C1 samples, while under-expressed in subtype C2 (Figure 8B). To elucidate the differences in the immune infiltration landscape among three subgroups, the CIBERSORT algorithm was used to infer immune cell infiltration and evaluate HLA and immune checkpoint gene sets expression. We also revealed significant differential expression in 19 immune checkpoint genes, including CD160, CD27, CD48, CD96, TNFRSF14, and TNFRSF17 (Figure 8C). Notably, these genes were significantly high expressed in the subtype C2 samples, except for B2M, HLA-A, HLA-C, IL6R, PDCD1, and PVR. And we also found that mast cells activated and monocytes in subtype C1 were remarkedly increased versus subtype C2 (P < 0.05) (Figure 8D). Compared with subtype C1, the infiltrating of Macrophages M1, T cells follicular helper, and T cells gamma delta were significantly increased in subtype C2 samples. These findings demonstrated that the immune infiltration landscape in RA samples is differentially affected by distinct expression patterns of DEGs associated with cellular senescence. According to your comments, we have revised the manuscript and Figure 8 as follows:

Page 11-12, section “Immune infiltration landscape of distinct subtypes”

“Through consensus clustering analysis, RA samples were grouped into two subgroups on the basis of the 29 cellular senescence-related DEGs mRNA expression, namely, subtype C1 (n = 12), subtype C2 (n = 21) (Figure 8A). Our results showed that BLVRA, ASF1A, MATK, and TLR3 tended to be under-expressed in subtype C1 samples, while high-expressed in subtype C2 samples (Figure 8B). We also found that genes SMARCA4, SIN3B, and MA2K7 were high-expressed in subtype C1 samples, while under-expressed in subtype C2 samples (Figure 8B). To elucidate the differences in the immune infiltration landscape among three subgroups, the CIBERSORT algorithm was used to infer immune cell infiltration and evaluate HLA and immune checkpoint gene sets expression. We also revealed significant differential expression in 19 immune checkpoint genes, including CD160, CD27, CD48, CD96, TNFRSF14, and TNFRSF17 (Figure 8C). Notably, these genes were significantly high expressed in the subtype C2 samples, except for B2M, HLA-A, HLA-C, IL6R, PDCD1, and PVR. And we also found that mast cells activated and monocytes in subtype C1 were remarkedly increased versus subtype C2 (P < 0.05) (Figure 8D). Compared with subtype C1, the infiltrating of Macrophages M1, T cells follicular helper, and T cells gamma delta were significantly increased in subtype C2 samples. ”

8. the study also lacks validation from basic experiments.

REPLY: Thank you for your comments. We also understand that our study lacks basic experiments for verification, but due to limited funds, we do not have the ability and economy to do basic experiments. To further improve our manuscript, we have added analysis of targeted drugs associated signature genes. To further explore whether the signature genes could be used as potential drug targets, we first using the “pRRophetic” R package to estimate the sample's drug sensitivity IC50 value for 18 drugs from CCLE database. Subsequently, Spearman's rank correlation analysis was performed to asses the association between the expression of signature genes and drug sensitivity. The results showed that the expression of DHX9 was significantly negative correlation with the sensitivity of PLX4720, PD.0325901, Axitinib, and Bosutinib (Figure 9A). The expression of CYR61 was significantly negative correlation with the sensitivity of Axitinib, and positive correlation with the sensitivity of PF2341066, PD.0332991, Cisplatin (Figure 9A). The expression of ITGB4 was significantly negative correlation with the sensitivity of AZD.0530, and Dasatinib (Figure 9A). We first searched the Drugbank database for RA treatment drugs, ibuprofen and indometacin are available for RA treatment. The results of AutoDock displayed that the binding energy of gene CYR61 and ibuprofen is -6.03 kcal/mol (Figure 9B), while the binding energy of gene CYR61 and indometacin is -6.9 kcal/mol (Figure 9C). Moreover, the binding energy of gene DHX9 and ibuprofen is -4.07 kcal/mol (Figure 9D), while the binding energy of gene DHX9 and indometacin is -5.96 kcal/mol (Figure 9E). There findings suggested that the signature genes (CYR61 and DHX9) could be used as potential drug targets for RA treatment.

According to your comments, we have added the analysis of targeted drugs associated signature genes and Figure 9 to demonstrate our results. We also also revised our manuscript as follows:

Page 8, section “Association between signature genes and drug sensitivity”

“Association between signature genes and drug sensitivity

To further explore whether the signature genes could be used as potential drug targets, we first using the “pRRophetic” R package to estimate the sample's drug sensitivity IC50 value for 18 drugs from CCLE database. pRRophetic built a model based on the known cell line expression matrix and drug sensitivity information, then the new expression matrix is predicted. Subsequently, Spearman's rank correlation analysis was performed to asses the association between the expression of signature genes and drug sensitivity. Besides, we also searched the treatments for rheumatoid arthritis in Drugbank database. 

AutoDock is an approach to drug design through the characterisation of receptors and the mode of interaction between receptors and drug molecules. A theoretical simulation method that focuses on the study of intermolecular (e.g., ligand-receptor) interactions and predicts their binding modes and affinities. AutoDock is the process of simulating molecular recognition in a computer, with the aim of finding the optimal binding conformation of a protein and its ligand and ensuring that the overall binding free energy of the complex is minimised. To further explore whether the signature genes we identified could be potential targets for these retrieved rheumatoid arthritis drugs, AutoDock was performed.”

Page 13, section “Association between signature genes and drug sensitivity”

“Association between signature genes and drug sensitivity

To further explore whether the signature genes could be used as potential drug targets, we first using the “pRRophet

---

## [Decision Letter · Decision Letter 1]

11 Jul 2024

PONE-D-24-02627R1Cellular Senescence-Associated Genes in Rheumatoid Arthritis: Identification and Functional AnalysisPLOS ONE

Dear Dr. Zhang,

Thank you for submitting your manuscript to PLOS ONE. After careful consideration, we feel that it has merit but does not fully meet PLOS ONE’s publication criteria as it currently stands. Therefore, we invite you to submit a revised version of the manuscript that addresses the points raised during the review process.

We look forward to receiving your revised manuscript.

Kind regards,

Gurudeeban Selvaraj

Academic Editor

PLOS ONE

Reviewers' comments:

Reviewer's Responses to Questions

**Comments to the Author**

1. If the authors have adequately addressed your comments raised in a previous round of review and you feel that this manuscript is now acceptable for publication, you may indicate that here to bypass the “Comments to the Author” section, enter your conflict of interest statement in the “Confidential to Editor” section, and submit your "Accept" recommendation.

Reviewer #1: All comments have been addressed

Reviewer #2: (No Response)

2. Is the manuscript technically sound, and do the data support the conclusions?

Reviewer #1: Yes

Reviewer #2: Partly

3. Has the statistical analysis been performed appropriately and rigorously? 

Reviewer #1: Yes

Reviewer #2: Yes

4. Have the authors made all data underlying the findings in their manuscript fully available?

Reviewer #1: Yes

Reviewer #2: No

5. Is the manuscript presented in an intelligible fashion and written in standard English?

Reviewer #1: Yes

Reviewer #2: Yes

6. Review Comments to the Author

Reviewer #1: (No Response)

Reviewer #2: Dear Editor and Authors,

Overall Assessment

I highly value the quality and findings presented in the manuscript and its intentions, objectives, and methodology. The strategy followed is appropriate, and the manuscript is worthy of publication in PLOS One. However, several aspects require significant major revision to strengthen the study further.

Methodology of Random Forest

While the use of Random Forest methodology has been relatively successful in developing a decision network, it has certain limitations that should be explored within the context of the study. Even though the ROC results presented are excellent, it is essential to address the possibility that the set of 29 genes identified may be incomplete and some relevant genes might have been overlooked. In this regard, comparing the predictive capabilities with an XGBoost model would be beneficial. XGBoost allows for monitoring and learning according to a loss function, potentially offering better performance in identifying additional relevant genes. Additionally, XGBoost could enhance the detection of the 29 previously selected genes by providing:

Enhanced Sensitivity and Specificity:

By optimizing function loss more effectively, XGBoost could potentially improve the sensitivity and specificity of gene detection, ensuring that more relevant genes are identified without increasing false positives.

Better Handling of Feature Interactions:

XGBoost can more effectively capture complex interactions between genes than Random Forest. This could lead to a more accurate identification of the key genes involved in the disease process.

Increased Robustness:

Through its regularization parameters, XGBoost reduces the risk of overfitting, ensuring that the identified genes are relevant and not artifacts of the training data.

Docking Studies

The proposed docking studies are a good starting point, but I identify several needs that are important to address:

Reference to AutoDock Procedure:

Appropriately referencing the AutoDock procedure is essential. Extensive, updated, and relevant literature on this topic exists, and a proper reference reflecting the use of the chosen suite is necessary.

Protein Structures:

The protein structures encoding the selected genes were obtained from PDB and possibly other sources. Given the advantages and excellent results provided by AlphaFold, I recommend that the authors compare the secondary structure of the proteins encoded by these genes with those found in PDB using AlphaFold predictions. This could offer additional validation and improve structural accuracy.

Topological Parameters of Ligands

Although the selected ligands are biochemically appropriate, it would be interesting to detail how the topological parameters of these ligands were obtained. If these parameters were resolved using DFT (Density Functional Theory), it is essential to specify which bases or functionals were used. Alternatively, the specific force field used and the rationale behind its selection should be explained if force fields were employed.

Molecular Dynamics Simulations:

While the AutoDock results are appropriate and relatively solid, I recommend that the authors conduct additional molecular dynamics simulations. Using an effective platform such as GROMACS for these simulations could provide valuable insights into the stability of the formed complex. Molecular dynamics simulations would allow for observing how the docking complex behaves in a simulated environment and verifying its stability over time.

Conclusion

The manuscript presents significant findings and utilizes robust methodologies, but it would benefit from the proposed improvements and major revisions. Implementing these recommendations could strengthen the study's results and conclusions, providing additional validation and a deeper understanding of the underlying biological mechanisms.

7. PLOS authors have the option to publish the peer review history of their article (what does this mean?). If published, this will include your full peer review and any attached files.

Reviewer #1: **Yes: **Ji Yin

Reviewer #2: **Yes: **Vicente Domínguez-Arca

---

## [Author Response · Author response to Decision Letter 1]

23 Aug 2024

Dear Prof. Gurudeeban Selvaraj and reviewers:

Thank you very much for giving us another opportunity to revise and resubmit our manuscript, we appreciate editor and reviewers very much for their positive and constructive comments and suggestions on our manuscript entitled “Cellular Senescence-Associated Genes in Rheumatoid Arthritis: Identification and Functional Analysis” (PONE-D-24-02627). Those comments are all valuable and helpful for improving our paper. We have studied reviewer’s comment carefully and have made revision which marked in red in the paper. And we have tried our best to revise our manuscript according to the comments. The main corrections in the paper and the response to the reviewers’ comments are as follows:

Comments to the Author

4. Have the authors made all data underlying the findings in their manuscript fully available?

Reviewer #1: Yes

Reviewer #2: No

REPLY: Thank you very much for your comments. In our manuscript, five rheumatoid arthritis datasets were used, including GSE55457, GSE12021, GSE55235, GSE77298, and GSE178557. And all the datasets were downloaded from GEO database. The comprehensive details of these five cohorts are delineated in Table 1 of our study. In parallel, the specifics of these genes are cataloged in Supplementary Table S1 of our documentation. That’s to say, we have made all data underlying the findings in our manuscript fully available. If the reviewer thinks that we still need to provide more detailed information, we will be glad to do so.

6. Review Comments to the Author 

Reviewer #1: (No Response) 

Reviewer #2: Dear Editor and Authors, 

Overall Assessment 

I highly value the quality and findings presented in the manuscript and its intentions, objectives, and methodology. The strategy followed is appropriate, and the manuscript is worthy of publication in PLOS One. However, several aspects require significant major revision to strengthen the study further. 

REPLY: Thank you very much for your further modification comments. We have studied your comment carefully and have made revision to meet PLOS ONE’s publication criteria.

Methodology of Random Forest

While the use of Random Forest methodology has been relatively successful in developing adecision network, it has certain limitations that should be explored within the context of the study. Even though the ROC results presented are excellent, it is essential to address the possibility that the set of 29 genes identified may be incomplete and some relevant genes might have been overlooked. In this regard, comparing the predictive capabilities with an XGBoost model would be beneficial. XGBoost allows for monitoring and learning according to a loss function, potentially offering better performance in identifying additional relevant genes. Additionally, XGBoost could enhance the detection of the 29 previously selected genes by providing:

Enhanced Sensitivity and Specificity:

By optimizing function loss more effectively, XGBoost could potentially improve the sensitivity and specificity of gene detection, ensuring that more relevant genes are identified without increasing false positives.

Better Handling of Feature Interactions:

XGBoost can more effectively capture complex interactions between genes than Random Forest. This could lead to a more accurate identification of the key genes involved in the disease process.

Increased Robustness:

Through its regularization parameters, XGBoost reduces the risk of overfitting, ensuring that the identified genes are relevant and not artifacts of the training data.

REPLY: Thank you very much for your further modification comments. In our manuscript, differential expressed analysis was first carried out between RA and normal samples. In total, there were 2050 DEGs identified using the filtering criteria of P-value less than 0.05 and the absolute value of log2 FC greater than 0.5 (Figure 2A). Next, we investigated the differential expression of cellular senescence-related genes and found that 29 of the 279 cellular senescence-related genes were significantly differentially expressed (Figure 2B). Subsequently, the random forest algorithm and LASSO regression analysis were respectively performed. The results showed that there were 3 cellular senescence-related DEGs identified by random forest algorithm and LASSO regression analysis respectively (Figure 2C-D). Then, 3 out of the 29 cellular senescence-related DEGs were ultimately remained as hub genes, including DHX9, CYR61 and ITGB (Figure 2C-D). According to your suggestions, we also performed XGBoost based on the expression levels of 29 cellular senescence-related genes after differential expression analysis. And our analysis found that there are 12 genes remained through XGBoost algorithm, but through LASSO regression analysis, there are still 3 cellular senescence-related DEGs retained, including DHX9, CYR61 and ITGB. This result shows that although XGBoost algorithm can retain more relevant genes, after LASSO algorithm, only three genes are still retained for model construction, which is consistent with the result of random forest. On the other hand, the purpose of our study is to detect fewer signature genes to distinguish between RA and normal samples, thus reducing the detection cost. Therefore, in this study, we are inclined to use random forest to further screen cellular senescence related genes that are more relevant to RA.

Docking Studies

The proposed docking studies are a good starting point, but I identify several needs that are important to address:

Reference to AutoDock Procedure:

Appropriately referencing the AutoDock procedure is essential. Extensive, updated, and relevant literature on this topic exists, and a proper reference reflecting the use of the chosen suite is necessary.

REPLY: Thank you very much for your further modification comments. According to your suggestions, we have appropriately reference the AutoDock procedure in our manuscript, and we have revised the manuscript as follows:

Page 8-9, section “Association between signature genes and drug sensitivity” in Materials and Methods

“To further explore whether the signature genes we identified could be potential targets for these retrieved rheumatoid arthritis drugs, AutoDock was performed. Detailedly, the crystal structures of signature genes were downloaded from RCSB Protein Data Bank (PDB, https://www.rcsb.org/) and the 3D structure of small molecule compound was downloaded from the PubChem Compound database. The downloaded complex were embellished by PyMol2.3.0 to remove original water molecules and phosphates. Moreover, the AutoDock Tools 4.2.6 [PMID: 19399780] (https://autodock.scripps.edu/) was used to prepare receptors, including adding Gasteiger charges, merging non-polar hydrogen bonds and setting docking parameters. A box of 60*60*60-point grid (0.375-Å spacing between the grid points) and the affinity maps were generated and processed using AutoGrid 4.2 by default setting. For each docking case, 200 Lamarckian genetic algorithm runs were processed by default setting using AutoDock 4.2.6. To further confirm the binding of small molecule compound to the selected genes, we have also obtained the structures of proteins encoded by the selected genes from AlphaFold and then re-performed AutoDock. The top-scored hit was chosen and visualized for further analysis and the PyMOL was used to preparse all the molecular graphics.”

Protein Structures:

The protein structures encoding the selected genes were obtained from PDB and possibly other sources. Given the advantages and excellent results provided by AlphaFold, I recommend that the authors compare the secondary structure of the proteins encoded by these genes with those found in PDB using AlphaFold predictions. This could offer additional validation and improve structural accuracy.

REPLY: Thank you for your constructive comments. According to your suggestions, to further confirm the binding of small molecule compound to the selected genes, we have also obtained the structures of proteins encoded by the selected genes from AlphaFold and then re-performed AutoDock. As shown by the docking results, ibuprofen and indometacin both possessed high binding affinity with an average of -7.25 kcal/mol. Specifically, the binding energy of gene CYR61 and ibuprofen is -6.55 kcal/mol (Supplementary Figure S2A), while the binding energy of gene CYR61 and indometacin is -9.01 kcal/mol (Supplementary Figure S2B). Moreover, the binding energy of gene DHX9 and ibuprofen is -6.43 kcal/mol (Supplementary Figure S2C), while the binding energy of gene DHX9 and indometacin is -7.02 kcal/mol (Supplementary Figure S2D). There findings offered additional validation that the signature genes (CYR61 and DHX9) could be used as potential drug targets for RA treatment.

We have revised the manuscript as follows:

Page 8-9, section “Association between signature genes and drug sensitivity” in Materials and Methods

“To further explore whether the signature genes we identified could be potential targets for these retrieved rheumatoid arthritis drugs, AutoDock was performed. Detailedly, the crystal structures of signature genes were downloaded from RCSB Protein Data Bank (PDB, https://www.rcsb.org/) and the 3D structure of small molecule compound was downloaded from the PubChem Compound database. The downloaded complex were embellished by PyMol2.3.0 to remove original water molecules and phosphates. Moreover, the AutoDock Tools 4.2.6 [PMID: 19399780] (https://autodock.scripps.edu/) was used to prepare receptors, including adding Gasteiger charges, merging non-polar hydrogen bonds and setting docking parameters. A box of 60*60*60-point grid (0.375-Å spacing between the grid points) and the affinity maps were generated and processed using AutoGrid 4.2 by default setting. For each docking case, 200 Lamarckian genetic algorithm runs were processed by default setting using AutoDock 4.2.6. To further confirm the binding of small molecule compound to the selected genes, we have also obtained the structures of proteins encoded by the selected genes from AlphaFold and then re-performed AutoDock. The top-scored hit was chosen and visualized for further analysis and the PyMOL was used to preparse all the molecular graphics.”

Page 14-15, section “Association between signature genes and drug sensitivity” in Results

“To further confirm the binding of small molecule compound to the selected genes, we have also obtained the structures of proteins encoded by the selected genes from AlphaFold and then re-performed AutoDock. As shown by the docking results, ibuprofen and indometacin both possessed high binding affinity with an average of -7.25 kcal/mol. Specifically, the binding energy of gene CYR61 and ibuprofen is -6.55 kcal/mol (Supplementary Figure S2A), while the binding energy of gene CYR61 and indometacin is -9.01 kcal/mol (Supplementary Figure S2B). Moreover, the binding energy of gene DHX9 and ibuprofen is -6.43 kcal/mol (Supplementary Figure S2C), while the binding energy of gene DHX9 and indometacin is -7.02 kcal/mol (Supplementary Figure S2D). There findings suggested that the signature genes (CYR61 and DHX9) could be used as potential drug targets for RA treatment.”

Topological Parameters of Ligands

Although the selected ligands are biochemically appropriate, it would be interesting to detail how the topological parameters of these ligands were obtained. If these parameters were resolved using DFT (Density Functional Theory), it is essential to specify which bases or functionals were used. Alternatively, the specific force field used and the rationale behind its selection should be explained if force fields were employed.

REPLY: We appreciate the reviewer’s insightful suggestion to elaborate on the determination of the topological parameters of the selected ligands. In our study, these parameters were derived using the AutoDock suite, specifically employing AutoDockTools, which are optimized for ligand preparation in docking simulations. Detailed Ligand Preparation and Parameterization as follows:

Hydrogen Addition: All ligand structures were first processed to ensure that hydrogen atoms were correctly added. This step is crucial for accurate representation of the ligand’s geometry and for subsequent charge calculations. 

Charge Assignment: Gasteiger charges were computed for all ligand atoms. The Gasteiger method, implemented in AutoDockTools, is a well-established approach for assigning partial atomic charges, which are critical for the electrostatic component of the docking scoring function. 

Atom Typing: Atom types were assigned according to the AutoDock parameterization scheme. This ensures that all atoms in the ligands are compatible with the scoring functions used by AutoDock. 

And the selection of AutoDockTools for ligand parameterization was based on several key considerations: 

Integration and Compatibility: AutoDockTools is specifically designed to integrate seamlessly with the AutoDock docking engine, ensuring that all topological parameters are fully compatible with the scoring functions and algorithms used in the docking simulations. 

Optimization for Docking: The parameterization methods in AutoDockTools, such as the Gasteiger charge assignment and specific atom typing, are optimized for the accurate prediction of ligand-receptor interactions within the AutoDock framework.

Efficiency: AutoDockTools provides an automated and efficient workflow for ligand preparation, which is particularly advantageous for handling multiple ligands in high-throughput docking studies.

Justification Against DFT and Force Fields

While Density Functional Theory (DFT) and specific force fields offer high precision in topological parameterization, the practical considerations of our study necessitated the use of AutoDockTools:

DFT Calculations: Performing DFT calculations to determine topological parameters for all ligands would be computationally intensive and time-prohibitive, given the scope of our study. Additionally, integrating DFT-derived parameters with the AutoDock scoring function could introduce compatibility issues.

Force Fields: Similarly, the use of external force fields would require additional parameter conversion steps and might not align perfectly with the AutoDock scoring function. The AutoDockTools parameterization is specifically tailored to the AutoDock scoring functions, ensuring consistency and reliability.

Utilizing AutoDockTools for ligand parameterization provided a streamlined and consistent approach that is directly aligned with the requirements and optimizations of the AutoDock suite. This methodology ensured reliable and reproducible docking results, forming a robust basis for our study’s conclusions. We hope this detailed explanation addresses the reviewer's concerns and clarifies our methodology. We are grateful for the opportunity to improve our manuscript based on the reviewer’s valuable feedback.

Molecular Dynamics Simulations:

While the AutoDock results are appropriate and relatively solid, I recommend that the authors conduct additional molecular dynamics simulations. Using an effective platform such as GROMACS for these simulations could provide valuable insights into the stability of the formed complex. Molecular dynamics simulations would allow for observing how the docking 

---

## [Decision Letter · Decision Letter 2]

17 Dec 2024

PONE-D-24-02627R2Cellular Senescence-Associated Genes in Rheumatoid Arthritis: Identification and Functional AnalysisPLOS ONE

Dear Dr. Zhang,

Thank you for submitting your manuscript to PLOS ONE. After careful consideration, we feel that it has merit but does not fully meet PLOS ONE’s publication criteria as it currently stands. Therefore, we invite you to submit a revised version of the manuscript that addresses the points raised during the review process.

We look forward to receiving your revised manuscript.

Kind regards,

Gurudeeban Selvaraj, PhD

Academic Editor

PLOS ONE

Journal Requirements:

Reviewers' comments:

Reviewer's Responses to Questions

**Comments to the Author**

1. If the authors have adequately addressed your comments raised in a previous round of review and you feel that this manuscript is now acceptable for publication, you may indicate that here to bypass the “Comments to the Author” section, enter your conflict of interest statement in the “Confidential to Editor” section, and submit your "Accept" recommendation.

Reviewer #2: All comments have been addressed

2. Is the manuscript technically sound, and do the data support the conclusions?

Reviewer #2: Yes

3. Has the statistical analysis been performed appropriately and rigorously? 

Reviewer #2: Yes

4. Have the authors made all data underlying the findings in their manuscript fully available?

Reviewer #2: Yes

5. Is the manuscript presented in an intelligible fashion and written in standard English?

Reviewer #2: Yes

6. Review Comments to the Author

Reviewer #2: Dear Authors,

First of all, I apologize for the delay in responding with my opinion on the review carried out by the authors. After reviewing the additions, based on the suggestions I previously made, some aspects of the article have been clarified, especially those related to the analysis conditions related to docking. Regarding my considerations on the possible improvements related to the use of an XGBoost classification architecture instead of RF, after thinking about it in detail, with the new explanations and comparisons with tests carried out with both architectures, I agree with the authors that the RF-based strategy seems to be the most appropriate in this case. Even so, I would like to point out that although minimizing the number of genes involved indeed translates into a potentially lower cost and "simplicity," taking into account the importance of these two factors, I think that it may be necessary to also mention the negative impact that this excessive minimization of genes involved can have on sensitivity and/or specificity, basically to fine-tune the elimination of possible false negatives. Thus, although perhaps a trade-off analysis could have been performed to see how sensitivity and specificity are affected as the number of selected genes is reduced, I understand that this task may be outside the clear objective presented in this incipient article, although perhaps it could be kept in mind for the future evolution of the work. In this sense, I think that perhaps reference could be made to this possible debate in the manuscript so that the need to examine in detail the compromise between the reduction of the number of genes involved and the possible effect on the sensitivity and specificity of the technique is made clear. Perhaps the use of XGBoost and the greater number of genes with detected involvement could be mentioned, although the three underlined ones are used in the end.

I appreciate that the information and details of the molecular structures in their bioinformatics version have been substantially increased, mentioning different databases and approximations that are used, such as the addition of hydrogens or the type of partial charges.

Finally, I appreciate that you have understood the good intentions and probabilities of significant findings using molecular dynamics results, and I fully understand the timeline and the specific purpose of the manuscript. I understand that it falls within the limits of this manuscript, but I consider it necessary to name docking as a possible evolution since there are very relevant issues, such as interaction through hydrogen bonds, that may be very poorly represented in docking and would require a detailed analysis of molecular dynamics to study in detail both the affinity and the binding and their mechanisms. In this sense, I consider it appropriate to mention it so that readers are aware of the need to answer some questions. All of this, of course, without implying a level of detail that conflicts with future work that the authors are planning or carrying out.

With these minor revisions that I propose, as far as my opinion is concerned, I consider that the article very pertinently complies with an originality and impact suitable to be presented to the scientific community. I thank you for taking my opinions into account and for giving me the opportunity to participate in the review of this valuable manuscript.

7. PLOS authors have the option to publish the peer review history of their article (what does this mean?). If published, this will include your full peer review and any attached files.

Reviewer #2: **Yes: **Vicente Domínguez-Arca

---

## [Author Response · Author response to Decision Letter 2]

24 Dec 2024

Subject: Re: Manuscript PONE-D-24-02627R2

Dear Dr. Selvaraj,

Thank you for providing the opportunity to revise our manuscript titled “Cellular Senescence-Associated Genes in Rheumatoid Arthritis: Identification and Functional Analysis.” We appreciate the reviewers’ and editor’s constructive comments, which have significantly improved our work. We have carefully addressed all points raised, and a detailed point-by-point response is attached. The revisions include:

Ensuring data availability as per PLOS ONE guidelines.

Updating the reference list for completeness and accuracy.

Please find the following documents submitted for your consideration:

A rebuttal letter detailing our responses.

A revised manuscript with changes tracked.

A clean manuscript version without tracked changes.

Thank you again for considering our manuscript, and we look forward to your feedback on this revised submission.

Response to Reviewer #2:

Dear Reviewer,

Thank you for your detailed and thoughtful feedback on our manuscript. We appreciate your acknowledgment of the clarifications and improvements we have made. Below, we outline how we have addressed your suggestions:

Trade-Off Between Gene Reduction and Model Performance:

We have incorporated a discussion on the potential trade-offs between reducing the number of selected genes and the impact on sensitivity and specificity. While this analysis is outside the current scope, we agree it is an important consideration for future work. The revised manuscript as follows:

Page 16-17, section “Discussion”

“We acknowledge that minimizing the number of selected genes can potentially reduce costs and improve interpretability. However, this reduction might also negatively impact sensitivity and specificity, possibly increasing the risk of false negatives. Future studies could explore this trade-off in greater detail, performing comprehensive analyses to evaluate sensitivity and specificity as a function of gene selection thresholds. Furthermore, while we chose an RF-based strategy for its robustness in this context, exploring XGBoost or other advanced machine learning architectures in future studies could provide additional insights into the potential contribution of a larger gene set.”

Mention of Molecular Dynamics:

In response to your suggestion, we have added a brief discussion highlighting the limitations of docking and the potential for molecular dynamics to provide deeper insights into molecular interactions, particularly hydrogen bonding. The revised manuscript as follows:

Page 14, section “Association between signature genes and drug sensitivity”

“Molecular docking (AutoDock) provides a computationally efficient approach to evaluate molecular interactions; however, it may inadequately capture certain specific interactions, such as hydrogen bonding. Molecular dynamics simulations offer a more comprehensive framework for understanding binding affinities and mechanisms. In our future studies, we could leverage these simulations to complement and refine the insights gained from docking analyses.”

Methodological Details:

We have ensured that the additional details regarding databases, hydrogen addition, and charge types are clearly described in the methodology section. The revised manuscript as follows:

Page 9, section “Association between signature genes and drug sensitivity” in Materials and Methods

“Detailedly, the crystal structures of signature genes were downloaded from RCSB Protein Data Bank (PDB, https://www.rcsb.org/) [PMID: 24597646] and the 3D structure of small molecule compound was downloaded from the PubChem Compound database. The downloaded complex was embellished by PyMol2.3.0 to remove original water molecules and phosphates. Moreover, the AutoDock Tools 4.2.6 [PMID: 19399780] (https://autodock.scripps.edu/) was used to prepare receptors, including adding Gasteiger charges, merging non-polar hydrogen bonds and setting docking parameters. A box of 60*60*60-point grid (0.375-Å spacing between the grid points) and the affinity maps were generated and processed using AutoGrid 4.2 by default setting. For each docking case, 200 Lamarckian genetic algorithm runs were processed by default setting using AutoDock 4.2.6. To further confirm the binding of small molecule compound to the selected genes, we have also obtained the structures of proteins encoded by the selected genes from AlphaFold and then re-performed AutoDock. The top-scored hit was chosen and visualized for further analysis and the PyMOL was used to preparse all the molecular graphics.”

We sincerely appreciate your constructive comments, which have improved the quality of our manuscript. Thank you for your valuable input.

Sincerely,

You Ao

Department of Orthopaedics, The fifth hospital of Harbin, Harbin, Heilongjiang, P.R. China

---

## [Editor Report · Decision Letter 3]

27 Dec 2024

Cellular Senescence-Associated Genes in Rheumatoid Arthritis: Identification and Functional Analysis

PONE-D-24-02627R3

Dear Dr. Zhang,

We’re pleased to inform you that your manuscript has been judged scientifically suitable for publication and will be formally accepted for publication once it meets all outstanding technical requirements.

Kind regards,

Zhanzhan Li

Academic Editor

PLOS ONE

Additional Editor Comments (optional):

The authors have addressed all commnets.

Reviewers' comments:

Na

---

## [Editor Report · Acceptance letter]

7 Jan 2025

PONE-D-24-02627R3 

PLOS ONE

Dear Dr. Zhang, 

I'm pleased to inform you that your manuscript has been deemed suitable for publication in PLOS ONE. Congratulations! Your manuscript is now being handed over to our production team.

Kind regards, 

on behalf of

Dr. Zhanzhan Li 

Academic Editor

PLOS ONE